# Diversity Enhanced Active Learning with Strictly Proper Scoring Rules

**Wei Tan**
Monash University
`wei.tan2@monash.edu`

Lan Du*
Monash University
`lan.du@monash.edu`

Wray Buntine
Monash University
`wray.buntine@monash.edu`

## Abstract

We study acquisition functions for active learning (AL) for text classification. The Expected Loss Reduction (ELR) method focuses on a Bayesian estimate of the reduction in classification error, recently updated with Mean Objective Cost of Uncertainty (MOCU). We convert the ELR framework to estimate the increase in (strictly proper) scores like log probability or negative mean square error, which we call Bayesian Estimate of Mean Proper Scores (BEMPS[2]). We also prove convergence results borrowing techniques used with MOCU. In order to allow better experimentation with the new acquisition functions, we develop a complementary batch AL algorithm, which encourages diversity in the vector of expected changes in scores for unlabelled data. To allow high performance text classifiers, we combine ensembling and dynamic validation set construction on pretrained language models. Extensive experimental evaluation then explores how these different acquisition functions perform. The results show that the use of mean square error and log probability with BEMPS yields robust acquisition functions, which consistently outperform the others tested.

## 1  Introduction

Classification has extensive uses and deep learning has substantially improved its performance, but a major hurdle for its use is the paucity of labelled or annotated data. The data labelling process performed by domain experts is expensive and tedious to produce, especially in the medical field, where due to lack of expertise and privacy issues, annotation costs are time-consuming and expensive [11]. Active Learning (AL) is an approach to speeding up learning by judiciously selecting data to be annotated [27]. AL is perhaps the simplest of all human-in-the-loop learning approaches, because the only interaction is the (expert) human providing a class label, yet for even in the simplest of tasks, classification, a general theory of AL is not agreed on.

Moreover, in practical situations, one needs to consider many issues when designing an AL system with deep learning: the expense to retrain deep neural networks and the use of validation data for training [31], transformer language models [16], batch mode AL with diversity [22], and consideration of expert capabilities and costs [8, 43]. Also important in experimental work is the need for realistic labelling sizes, with practitioners we work with saying expert annotation may allow a budget of up to 1000 data points, rarely more. Using large batch sizes (e.g., 1000 in [1, 20]) can thus be impractical.

Our fundamental research contribution is to suggest what makes a good acquisition function for the uncertainty component, without any batching. While there are many recent methods looking at the uncertainty diversity trade-off for batch AL [22], few recently have focused on understanding the uncertainty side alone. A substantial advance is a recent theoretical framework, Mean Objective

---

*Corresponding author

[2]Our implementation of BEMPS can be downloaded from `https://github.com/davidtw999/BEMPS`.

35th Conference on Neural Information Processing Systems (NeurIPS 2021).

Cost of Uncertainty (MOCU) [42] that provides a convergence proof. As our first contribution, we convert their expected loss reduction framework to an AL model using strictly proper scoring rules or Bregman divergences [9] instead of classification errors. MOCU required manipulations of the expected error, resulting in weighted-MOCU (WMOCU), in order to achieve convergence and avoid getting stuck in error bands. Strictly proper scoring rules naturally avoid this problem by generalising expected errors to expected scores. Using strictly proper scoring rules means better calibrated classifiers are rewarded. The scoring rules go beyond simple minimum errors of WMOCU and can be adapted to different kinds of inference tasks (e.g., different utilities, precision-recall trade-offs, etc.). This property is preferable and beneficial for applications such as medical domains where actual errors become less relevant for an inference task.

In order to evaluate the new acquisition functions we use text classification, which is our target application domain. For realistic evaluation, we want to use near state of the art systems, which means using pretrained language models with validation sets [24], and neural network ensembles [13]. Ensembling also doubles as a heuristic technique to yield estimates of model uncertainty and posterior probabilities. Coming up with a simple approach to combine ensembling and validations sets is our second research contribution. The importance of these combinations for AL has been noted [40, 25]. For further batch comparisons, we then bring back diversity into the research, suggesting a way to naturally complement our new family of acquisition functions with a method to achieve diversity, our third research contribution. Extensive experiments with a comprehensive set of ablation studies on four text classification datasets show that our BEMPS-based AL model consistently outperforms recent techniques like WMOCU and BADGE, although we explicitly exclude recent semi-supervised AL methods because they represent an unfair comparison against strictly supervised learning.

## 2   Related work

The proposed BEMPS, as a general Bayesian model for acquisition functions, quantifies the model uncertainty using the theory of (strictly) proper scoring rules for categorical variables [9]. Thus, we first review existing acquisition functions (aka query strategies) proposed in some recent AL models that are most related to ours, and refer interested readers to [22] for a comprehensive discussion of recent strategies. One common and simple heuristic often used is maximum entropy [45, 44, 34, 7], where one chooses samples that maximise the predictive entropy. Bayesian AL by disagreement (BALD) [10] and its batch version (i.e., BatchBALD) [12] instead compute the mutual information between the model predictions and the model parameters, which indeed chooses samples that maximise the decrease in expected entropy [20]. Recently, inspired by the one-step-look-ahead strategy of ELR [23], Zhao et al. [42] extended MOCU [39] to WMOCU with a theoretical guarantee of convergence. But WMOCU only applies to errors which are less relevant for some inference tasks. BEMPS naturally extends those methods, focusing on Bayesian estimation of uncertainty related to the model's expected proper score.

Ensemble methods are used to obtain better uncertainty estimates with deep learning, including Monte-Carlo dropout (MC-dropout) [6, 12, 21] and deep ensembles [13]. More sophisticated techniques are being developed, for instance, MCMC, hybrid and deterministic approaches [36, 2]. However, plain ensembling remains a competitive and simple method for deep learning [2].

Query strategies considering just uncertainty do not always work well in a batch setting, due to the highly similar samples acquired in a batch [35, 22, 17]. To overcome this problem, there have been many AL methods that achieve batch diversity by acquiring samples that are both informative and diverse, such as [19, 38, 14, 1, 40, 43, 29]. BADGE [1] and ALPS [40] are the two recent AL methods focusing on batch diversity. BADGE uses gradient embeddings of unlabelled samples as inputs of $k$-MEANS++ to select a set of diverse samples, which relies on fine-tuning pretrained language models. Instead of using gradient embeddings, ALPS uses surprisal embeddings computed from the predictive word probabilities generated by a masked language model for cold-start AL. Whereas our BEMPS computes for each unlabelled sample a vector of expected change in the proper scores, relating directly to the performance.

Other recent batch AL approaches, WAAL [30] and VAAL [32] use semi-supervised learning, for instance WAAL's models are trained on the feature space obtained with the help of the unlabelled data. We exclude these methods from our comparison since they are semi-supervised learning algorithms. Both Core-set [26] and WAAL [30] also set up sophisticated non-Bayesian cost functions using

bounds to deal with the many unknown probabilities when trying to optimise for a batch, whereas the Bayesian formula of BEMPS can be directly estimated.

In experimental work, the use of a validation set to train deep learning models in active learning is not uncommon, the goal of which is to avoid over-fitting and achieve early-stopping. Some existing methods assume that there is a large validation set available a prior [1, 12, 7], but means the cost of labelling the set is not factored into the labelling budget. We argue that the availability of a separate validation set is impractical in real world AL scenarios. Although Yuan et al. [40] use fixed epochs to train the classifier without a validation set to save the cost, the classifier could either be under-fit or over-fit. We instead use a dynamic approach to generate alternative validation sets from the ever increasing labelled pool after each iteration.

## 3 Bayesian estimate of mean proper scores

We first review the general Bayesian model for acquisition functions that includes ELR, MOCU and BALD. We then develop Bayesian Estimate of Mean Proper Scores (BEMPS), a new uncertainty quantification framework with a theoretical foundation based on strictly proper scoring rules [9]. It naturally extends ELR and BALD, focusing on the Bayesian estimation of uncertainty related to the models expected performance.

Suppose models of our interest are parameterised by parameters $\theta \in \Theta$, $L$ indicates labelled data, probability of label $y$ for data $x$ is given by $\Pr(y \mid \theta, x)$, and $\Pr(\cdot \mid \theta, x)$ presents a vector of label probabilities. With a *fully conditional model*, the posterior of $\theta$ is unaffected by unlabelled data, which means $\Pr(\theta \mid L, U) = \Pr(\theta \mid L)$ for any unlabelled data $U$. Moreover, we assume without loss of generality that this model family is well behaved in a statistical sense, so the model is identifiable. The "cost" of the posterior $\Pr(\theta \mid L)$ can be measured by some functional $Q(\Pr(\theta \mid L))$, denoted $Q(L)$ for short, where $Q(L) \geq 0$ and $Q(L) = 0$ when some convergence objective has been achieved. For our model this is when $\Pr(\theta \mid L)$ has converged to a point mass at a single model. A suitable objective function is to measure the expected decrease in $Q(\cdot)$ due to acquiring the label for a data point $x$. The corresponding acquisition function for AL is formulated [10, Equation (1)] as

$$\Delta Q(x|L) = Q(L) - \mathcal{E}_{\Pr(y|L,x)} \left[ Q(L \cup \{(x,y)\}) \right] , \tag{1}$$

whereas for ELR the expression is split over an inequality sign [23, Equation (2)]. It estimates how much the cost is expected to reduce when a new data point $x$ is acquired. Since the true label for the new data $x$ is unknown a prior, we have to use expected posterior proportions from our model, $\Pr(y \mid L, x)$ to estimate the likely label. For BALD [10] using Shannon's entropy, $Q_I(L) = I(\Pr(\theta \mid L))$, which measures uncertainty in the parameter space and thus has no strong relationship to actual errors [42]. MOCU and ELR use a Bayesian regret given by the expected loss difference between the optimal Bayesian classifier and the optimal classifier:

$$Q_{MOCU}(L) = \mathcal{E}_{\Pr(x')} \left[ \min_{y'}(1 - \Pr(y' \mid L, x')) - \mathcal{E}_{\Pr(\theta|L)} \left[ \min_{y'}(1 - \Pr(y' \mid \theta, x')) \right] \right] . \tag{2}$$

WMOCU uses a weighting function defined by Eq (11) in [42] to have a more amenable definition of $\Delta Q(x|L)$ than the MOCU method. Although WMOCU guarantees $\Delta Q(x|L)$ converging to the optimal classifier (under minimum errors) according to the $Q(L)$ with the strictly concave function by Eq (15) in [42], the optimal approximation of the convergences can only be solved by controlling a hyperparameter of the weighting function manually. To allow theoretical guarantees of convergence under more general loss functions, we propose a different definition for $Q(L)$ based on strictly proper scoring rules.

### 3.1 Strictly Proper Scoring Rules for Active Learning

A scoring rule assesses the quality of probabilistic prediction of categorical variables, and is often used in training a classification algorithm. For a model $\Pr(y \mid \theta, x)$ with input data $x$ and if one observes label $y$, the score is given by a function $S(\Pr(\cdot \mid \theta, x), y)$. A strictly proper scoring rule has the behaviour that in the limit of infinite labelled data $L_n$, as $n \to \infty$, the average score $\frac{1}{n} \sum_{(x,y) \in L_n} S(\Pr(\cdot \mid \theta, x), y)$ has a unique maximum for $\theta$ at the "true" model (for our identifiable

model family). The Savage representation [9] states that a strictly proper scoring rule for categorical variables takes the form $S(q(\cdot), y) = G(q(\cdot)) + dG(q(\cdot))(\delta_y - q(\cdot))$ for a strictly convex function $G(\cdot)$ with subgradient $dG(\cdot)$. Note that the expectation of a scoring rule according to the supplied probability takes a simple form $\mathcal{E}_{q(y)}\left[S(q(\cdot), y)\right] = \sum_y q(y)S(q(\cdot), y) = G(q(\cdot))$.

With strictly proper scoring rules, we develop a generalised class of acquisition functions built using the posterior (i.e., w.r.t. $\Pr(\theta \mid L)$) expected difference between the score for the Bayes optimal classifier and the score for the "true" model. This is inherently Bayesian due to the use of $\Pr(\theta \mid L)$.

$$
\begin{align}
Q_S(L) &= \mathcal{E}_{\Pr(x)\Pr(\theta|L)}\left[\mathcal{E}_{\Pr(y|\theta,x)}\left[S(\Pr(\cdot \mid \theta, x), y) - S(\Pr(\cdot \mid L, x), y)\right]\right] \tag{3} \\
&= \mathcal{E}_{\Pr(x)\Pr(\theta|L)}\left[B(\Pr(\cdot \mid L, x), \Pr(\cdot \mid \theta, x))\right] \tag{4} \\
&= \mathcal{E}_{\Pr(x)}\left[\mathcal{E}_{\Pr(\theta|L)}\left[G(\Pr(\cdot \mid \theta, x))\right] - G(\Pr(\cdot \mid L, x))\right] \tag{5} \\
\Delta Q_S(x|L) &= \mathcal{E}_{\Pr(x')}\left[\mathcal{E}_{\Pr(y|L,x)}\left[G(\Pr(\cdot \mid L, (x,y), x'))\right] - G(\Pr(\cdot \mid L, x'))\right] \tag{6}
\end{align}
$$

The $Q_S(L)$ has three equivalent variations, one for an arbitrary strictly proper scoring rule $S(q(\cdot), y)$ (Eq (3)), one for a corresponding Bregman divergence $B(\cdot, \cdot)$ (Eq (4)) and the third for an arbitrary strictly convex function $G(\cdot)$ (Eq (5)). Their connections are given in [9].

It is noteworthy that the acquisition function $\Delta Q_S(x|L)$ defined in Eq (6) is in a general form, applicable to any strictly proper scoring function for categorical variables. For instance, using a logarithmic scoring rule, popular for deep neural networks, we have $S_{log}(q(\cdot), y) = \log q(y)$ and $G_{log}(q(\cdot)) = -I(q(\cdot))$. Using the squared error scoring rule, known as a Brier score, we have $S_{MSE}(q(\cdot), y) = -\sum_{\hat{y}}\left(q(\hat{y}) - 1_{y=\hat{y}}\right)^2$ and $G_{MSE}(q(\cdot)) = \sum_y q(y)^2 - 1$. Combining these $G(\cdot)$ functions with Equation (6) yields two acquisition functions for the different scoring rules. These, as well as the corresponding for BALD have some elegant properties, with proofs given in Appendix A.

**Lemma 1** (Properties of scoring). *In the context of a fully conditional classification model* $\Pr(y \mid \theta, x)$, *the* $Q_I(L)$, $Q_S(L)$, $\Delta Q_I(x|L)$, $\Delta Q_S(x|L)$ *as defined above are all non-negative.*

Moreover, they guarantee learning will converge to the "truth".

**Theorem 1** (Convergence of active learning). *We have a fully conditional classification model* $\Pr(y \mid \theta, x)$, *for* $\theta \in \Theta$ *with finite discrete classes* $y$ *and input features* $x$. *Moreover, there is a unique "true" model parameter* $\theta_r$ *with which the data is generated, the prior distribution* $p(\theta)$ *satisfies* $p(\theta_r) > 0$, *and the model is identifiable. Then for the AL algorithm defined by the acquisition functions defined above of* $\Delta Q_I(x|L)$ *or* $\Delta Q_S(x|L)$ *after being applied for* $n$ *steps gives labelled data* $L_n$, *then* $\lim_{n\to\infty} \Delta Q_I(x|L_n) = 0$ *and likewise for* $Q_S(\cdot)$. *Moreover,* $\lim_{n\to\infty} \Pr(\theta \mid L_n)$ *is a delta function at* $\theta = \theta_r$ *for data acquired by both* $\Delta Q_I(x|L)$ *or* $\Delta Q_S(x|L)$.

Finiteness and discreteness of $x$ is used to adapt results from [42] to show for all $x$ that $\Delta Q(x|L_n) \to 0$ as $n \to \infty$, not an issue since real data is always finite. Interestingly $\Delta Q_I(x|L)$, i.e., BALD, achieves convergence too, which occurs because the model is identifiable and fully conditional, during AL we are free to choose $x$ values that would distinguish different parameter values $\theta$. Full conditionality also supports BEMPS because it means any inherent bias in the AL selection is nullified with the use of the data distribution $\Pr(x)$. But it also means that the theory has not been shown to hold for semi-supervised learning algorithms, where full conditionality does not apply.

Compared with BALD, MOCU and WMOCU, the advantage of using strictly proper scoring rules in BEMPS is that they generalise expected errors to expected scores, which can be tailored for different inference tasks. BALD uses mutual information to score samples based on how their labels could inform the true model parameter distribution, which will be problematic if the uncertainty of model parameters has a reduced relationship to the classification performance. This is reflected by its poor performance in our experiments. MOCU however has convergence issues as ELR, as pointed out by Zhao et al. [42]. Even though WMOCU can overcome the convergence issues, it is limited to the minimisation of expected errors.

In contrast, measuring the quality of predictive distributions via rewarding calibrated predictive distributions, scoring rules are in favour of adaptability. For instance, scoring rules can be developed [9] for some different inference tasks, including Brier score, logarithmic score, etc. Many loss

**Algorithm 1** Estimating point-wise $\Delta Q(x|L, x')$ with Equation (6)

**Require:** unlabelled data point $x$, existing labelled data $L$, estimation point $x'$
**Require:** model/network ensemble $\Theta^E = \{\theta_1, ..., \theta_E\}$ built from labelled data $L$,
**Require:** strictly convex function $G(\cdot)$ taking as input a probability density over $y$
1: $Q = 0$
2: $qx(\cdot) = \sum_{\theta \in \Theta^E} \Pr(\theta \mid L)) \Pr(\cdot \mid \theta, x)$
3: **for** $y$ **do**
4:     $q(\cdot) = \sum_{\theta \in \Theta^E} \Pr(\theta \mid L, (x, y)) \Pr(\cdot \mid \theta, x')$
5:     $Q \mathrel{+}= qx(y)G(q(\cdot))$
6: $q(\cdot) = \sum_{\theta \in \Theta^E} \Pr(\theta \mid L) \Pr(\cdot \mid \theta, x')$
7: $Q \mathrel{-}= G(q(\cdot))$
8: **return** $Q$

**Algorithm 2** Estimate of $\operatorname{argmax}_{x \in U} \Delta Q(x|L)$

**Require:** unlabelled pool $U$, estimation pool $X$
1: **for** $x \in U$ **do**
2:     $Q_x = 0$
3:     **for** $x' \in X$ **do**
4:         $Q_x \mathrel{+}= \Delta Q(x|L, x')$
5: **return** $\operatorname{argmax}_{x \in U} Q_x$

**Algorithm 3** Finding a diverse batch

**Require:** unlabelled pool $U$, batch size $B$
**Require:** estimation pool $X$, top fraction $T$
1: $\forall_{x \in U} Q_x = 0$
2: **for** $x \in U, x' \in X$ **do**
3:     $Q_x \mathrel{+}= vec_{x,x'} = \Delta Q(x|L, x')$
4: $V \leftarrow topk(Q, T * |U|)$
5: $batch = \emptyset$
6: $centroids = k\text{-Means centers } (vec_{x \in V}, B)$
7: **for** $c \in centroids$ **do**
8:     $batch \mathrel{\cup}= \{\operatorname{argmin}_{x \in V} ||c - vec_x||\}$
9: **return** $batch$

functions used by neural networks, like cross-entropy loss, are indeed proper scoring rules [13]. In most cases we can create a particular model to match just about any Bregman divergence (e.g., minimum squared errors is a Gaussian). In practice we can use robust models (e.g., a Dirichlet-multinomial rather than a multinomial, a negative binomial rather than a Poisson, Cauchy rather than Gaussian) in our log probability. For the particular cases we used, minimising Brier score gets the probability right in a least squares sense, i.e., minimising the squared error between the predictive probability and the one-shot label representation, which pays less attention to very low probability events. Meanwhile, log probability gets the probability scales right, paying attention to all events.

## 3.2 Acquisition Algorithms with Enhanced Batch Diversity

Algorithm 2 gives an implementation of BEMPS for an arbitrary strictly convex function $G(\cdot)$, returning the data point with the best estimated measure. To work with a Bregman divergence or proper score, the corresponding strictly convex function $G(\cdot)$ should first be derived. When $G(\cdot)$ is negative entropy, we call this CoreLog and when $G(\cdot)$ is the sum of squares we call this CoreMSE, corresponding to the log probability or Brier scoring rules respectively. Algorithm 2 calls Algorithm 1 to get the estimation at test point $x'$, which implements the function inside $\mathcal{E}_{\Pr(x')}[\cdot]$ in Equation (6). Note $\Pr(\theta \mid L, (x, y))$ is computed from $\Pr(\theta \mid L)$ using Bayes theorem. Both Algorithms 2 and 3 use a fixed *estimation pool*, $X$, a fixed random subset of the initial unlabelled data used to estimate expected values $\mathcal{E}_{\Pr(x')}[\cdot]$. Algorithm 3 returns $B$ data points representing a batch with enhanced diversity: it first calls Algorithm 1 to get, for each data point $x$ in the unlabelled pool, the vector of expected changes in score values over the estimation pool. Thus, this vector conveys information about uncertainty directly related to the change in score due to the addition of $x$. While the gradient embedding used in [1] represents a sample's impact on the model, our vector represents a sample's impact on the mean proper score. Concurrently Algorithm 3 computes the estimate of $\Delta Q(x|L)$ for these same $x$s. The top $T\%$ of scoring data $x$ are then clustered with $k$-Means and a representative of each cluster closest to the cluster mean is returned. This $k$-Means selection process tends to generate a diverse batch of high-scoring samples. The intuition is that 1) only higher scoring data $x$ should appear in a batch; 2) those clusters capture the pattern of expected changes in score values deduced by samples in the unlabelled pool, where the samples with a similar mean change in score values are grouped together; 3) samples in the same cluster can affect the learning similarly, so should not co-occur in a batch.

# 4 Experiments

We demonstrate the efficacy of BEMPS via evaluating the performance of CoreMSE and CoreLog as its two examples on various benchmark datasets for text classification. These two acquisition functions were compared with recent techniques for AL.

Table 1: Datasets and the used language model

| Dataset | Unlabelled/Test sizes | # Classes | Lang. Model | Initial labelled size |
|---|---|---|---|---|
| AG NEWS | 120,000 / 7,600 | 4 | DistilBERT | 26 |
| PUBMED 20K RCT | 15,000 / 2,500 | 5 | DistilBERT | 26 |
| IMDB | 25,000 / 25,000 | 2 | DistilBERT | 26 |
| SST-5 | 8544 / 2210 | 5 | DistilBERT | 26 |

Table 2: The average running time (in seconds) per single acquisition iteration with unlabelled pools of varied sizes, which were generated from AG NEWS.

| # Unlabelled | CoreMSE | CoreLog | Max-Ent | BALD | MOCU | WMOCU | BADGE | ALPS | Rand |
|---|---|---|---|---|---|---|---|---|---|
| 10k | 65 | 65 | 61 | 61 | 65 | 65 | 76 | 62 | <1 |
| 25k | 163 | 163 | 159 | 159 | 163 | 163 | 323 | 283 | <1 |
| 50k | 326 | 326 | 322 | 322 | 326 | 327 | 884 | 573 | <1 |
| 100k | 654 | 654 | 649 | 650 | 658 | 659 | 2904 | 1299 | <1 |

**Datasets** We used *four benchmark text datasets* for three different classification tasks: topic classification, sentence classification, and sentiment analysis, as shown in Table 1. The AG NEWS for topic classification contains 120K texts of four balanced classes [41]. The PUBMED 20k was used for sentence classification [3], which contains about 20K medical abstracts with five categories. This dataset is imbalanced. For sentiment analysis, we used both the SST-5 and the IMDB datasets. SST-5 contains 11K sentences extracted from movie reviews with five imbalanced sentiment labels [33], and IMDB contains 50K movie reviews with two balanced classes [18].

**Baselines** We considered Max-Entropy [37], BALD [10], MOCU [42], WMOCU [42], BADGE [1] and ALPS [40] together with a random baseline. Max-Entropy and BALD are earlier uncertainty-based methods. MOCU, WMOCU, BADGE and ALPS are the four recent AL methods. Following their originally published algorithms, we re-implemented them for text classification tasks, with the same backbone classifier for fair comparisons. Note that the hypothetical labels used in BADGE were computed with ensembles.

**Experiment setup** We used a small and fast pretrained language model, DistilBERT [24] as the backbone classifier in our experiments. We fine-tuned DistilBERT on each dataset after each AL iteration with a random re-initialization [5], proven to improve the model performance over the use of incremental fine-tuning with the newly acquired samples [7]. The maximum sequence length was set to 128, and a maximum of 30 epochs was used in fine-tuning DistilBERT with early stopping [4]. We used AdamW [15] as the optimizer with learning rate 2e-5 and betas 0.9/0.999. Meanwhile, the initial training and validation split contain only 20 and 6 samples respectively.

Each AL method was run for five times with different random number seeds on each dataset. The batch size $B$ was set to {1, 5, 10, 50, 100}. To compute the predictive distributions $\Pr(\cdot \mid L, x)$ and $\Pr(\cdot \mid L, (x, y), x')$, we borrowed the idea of deep ensembles [13], where we trained five DistilBERTs with randomly generated train-validation splits (i.e., train/validation=70/30%) based on an incrementally augmented labelled pool. Thus each member of the ensemble has a different train/validation set, helping to diversify the ensemble. We call this split process *dynamic validation set* (aka *Dynamic VS*). With it, our implementation of CoreMSE and CoreLog does not rely on a separate large validation set. More detailed experimental settings are given in Appendix B.

All experiments were run on 8 Tesla 16GB V100 GPUs. For each AL method, we computed the average running time per one acquisition iteration as the total running time divided by the total number of iterations (i.e., 10). Table 2 summarizes as an example the running times of all the AL methods on AG NEWS with unlabelled pools of different size. The batch size $B$ was set to 50, top fraction $T$ to 50% and the size of X to 600. Except for BADGE and the random baseline, the runtimes of the other methods increase nearly linearly with the size of the unlabelled pool.

**Comparative performance metrics** We followed Ash et al. [1] to compute a pairwise comparison matrix but use a counting-based algorithm [28], as shown in the left of Figure 1. The rows and columns of the matrix correspond to the AL methods used in our experiments. Each entry represents the outcome of the comparison between method $i$ and method $j$ over all datasets (D). Let $C_{i,j,d} = 1$ when method $i$ beats method $j$ on dataset $d$, and 0 otherwise. Each cell value of the matrix is them computed as $C_{i,j} = \sum_d^D C_{i,j,d}$. To determine the value of $C_{i,j,d}$, we used a two-side paired $t$-test

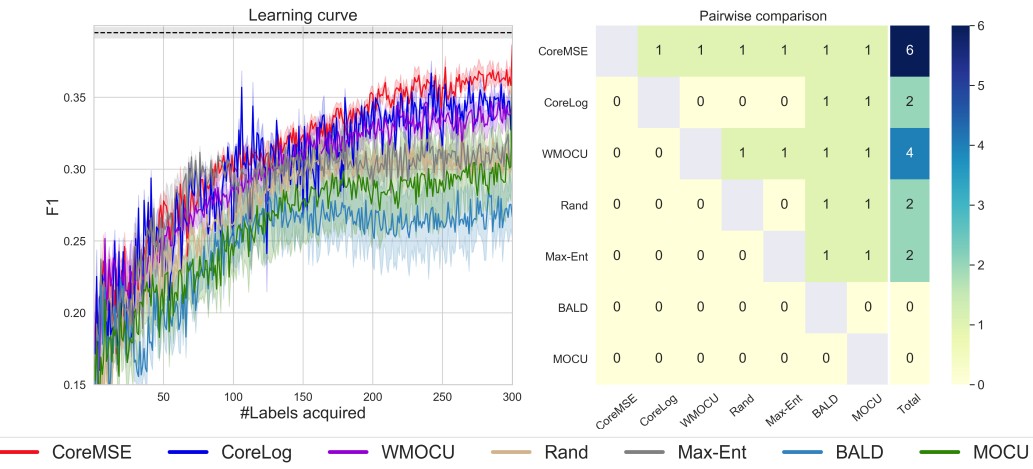

Figure 1: Performance on SST-5 dataset. The left half illustrates the learning curve, while the right half illustrates the matrix of paired comparisons. The dashline represents the performance of the backbone classifier trained on the entire dataset.

to compare their performance for 5 weighted F1 scores (or accuracy) at maximally spaced labelled sample sizes $\{l_{i,j,d}^1, l_{i,j,d}^2, ..., l_{i,j,d}^5\}$ from the learning curve. We compute the $t$-score as $t = \sqrt{5}\hat{\mu}/\hat{\sigma}$, where $\hat{\mu}$ and $\hat{\sigma}$ are the usual sample mean and std.dev. In Figure 1, we selected five samples at the different iterations according to the step size 50 in the experiment. For example, the first sample $l_{i,j,d}^1$ is chosen at the $50^{th}$ iteration, and the second sample $l_{i,j,d}^1$ is chosen at the $100^{th}$ iteration, etc. $\hat{\mu} = \frac{1}{5}\sum_{k=1}^5 \left(l_{i,j,d}^k\right), \hat{\sigma} = \sqrt{\frac{1}{4}\sum_{k=1}^5 \left(l_{i,j,d}^k - \hat{\mu}^2\right)}$. The $C_{i,j,d}$ is assigned to 1 if method $i$ beats method $j$ with $t$-score > 2.776 ($p$-value < 0.05). We accumulate the outcome of each pair comparison into the total quantity of each strategy (i.e., the "Total" column in the matrix). The highest total quantity gives a ranking over the AL methods. We also report the learning curves of all the AL methods with both weighted F1 score and accuracy.

## 4.1 Model performance: learning curves and comparative comparisons

**Active learning with batch size one** We first compared our CoreMSE and CoreLog based on Algorithm 2 to the baselines on the PUBMED and the SST-5 datasets to demonstrate how those methods perform particularly on a hard classification setting where classes are imbalanced. The learning curve sitting in the left of Figure 1 shows CoreMSE, CoreLog and WMOCU outperform all the other methods considered, we attribute to their estimation of uncertainty being better related to classification accuracy. Among these three methods, our CoreMSE performs the best in term of F1 score. The matrix at the right of Figure 1 then presents a statistical summary of comparative performance. CoreMSE has the highest total quantity which further confirms its effectiveness in acquiring informative samples in AL. More results on SST-5 and PUBMED, including both learning curves and comparative matrices are reported in Appendix C.

**Batch active learning** We compared batch CoreMSE and batch CoreLog implemented based on Algorithm 3 with BADGE, ALPS and batch WMOCU on the four datasets listed in Table 1, We extended WMOCU with our Algorithm 3 to build its batch counterpart. Specifically, we generated $vec_x$ using its point-wise error estimates, i.e., Eq (10) in [42]. The random baseline selected $B$ unlabelled samples randomly. Here we present the results derived with $B = 50$ as an example. More comprehensive results with different batch sizes, including accuracy, can be found in Appendix C.

The learning curves in Figure 2 show that batch CoreMSE and CoreLog almost always outperform the other AL methods as the number of acquired samples increases. Batch WMOCU devised with our batch algorithm compare favourably with BADGE and ALPS that use gradient/surprisal embeddings to increase batch diversity. These results suggest that selecting the representative samples from clusters learned with vectors of expected change in scores can better improve batch diversity, leading to an improved AL performance. Comparing CoreMSE/CoreLog with WMOCU further shows the advantage of BEMPS. Moreover, the performance differences between our methods and others

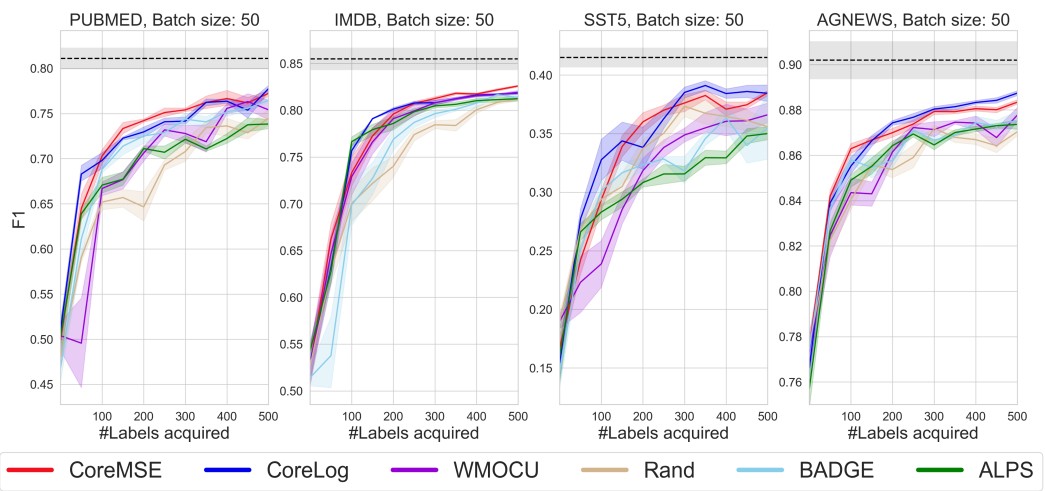

Figure 2: Learning curves of batch size 50 for PUBMED, IMDB, SST-5 and AG NEWS. The dashline represents the performance of the backbone classifier trained on the entire dataset.

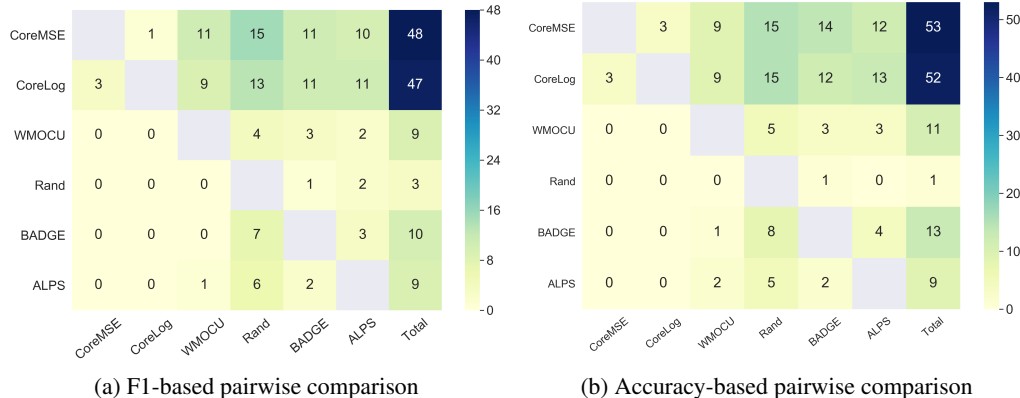

(a) F1-based pairwise comparison          (b) Accuracy-based pairwise comparison

Figure 3: Pairwise comparison matrices of batch active learning strategies.

on PUBMED and SST-5 indicate that batch CoreMSE and CoreLog can still achieve good results when the annotation budget is limited in those imbalanced datasets. We also created four pairwise comparison matrices for different batch sizes using either F1 score or accuracy. Figure 3 show the sum of the four matrices, summarizing the comparative performance on the four datasets. The maximum cell value is now $4 \times 4 = 16$. In other words, if a method beats another on all the four datasets across the four different batch sizes, the corresponding cell value will be 16. Both matrices computed with F1 score and accuracy respectively show both CoreMSE and CoreLog are ranked higher than the other methods. The observations discussed above are also consistent across different batch sizes.

## 4.2 Ablation Studies

**Batch size** In Figure 4, we plotted the learning curves of batch CoreMSE with different batch sizes (i.e., $B \in \{1, 5, 10, 50, 100\}$) on PUBMED and SST-5 as an example. The curves demonstrate that the performance of smaller batch sizes (5 or 10) is superior to that of large batch sizes (50 or 100), especially in the early rounds of training, but surprisingly also superior to that of the no-batch case. The no-batch case need to perform multiple one-step-look-ahead optimising acquisitions (Algorithm 1 and Algorithm 2) sequentially in order to acquire $B$ samples whereas the batch case only perform the acquisition at once, heuristically, incorporating a form of diversity based directly on the error surface (Algorithm 1 and Algorithm 3). This phenomena is also seen with BatchBALD [12, see Figure 4]. Thus we can conclude that the one-step-look-ahead of Equation (1) is only greedy, and not optimal. More results of Batch CoreMSE and CoreLog on PUBMED, SST-5, AG NEWS and IMDB can be found in Appendix B. In general, all the results shown in the learning curves prove that

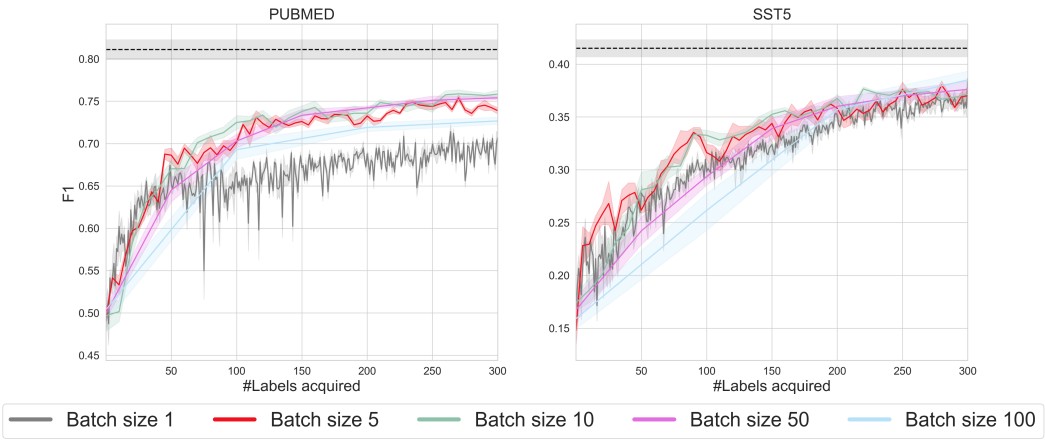

Figure 4: Learning curves of batch size 1, 5, 10, 50 and 100 for CoreMSE

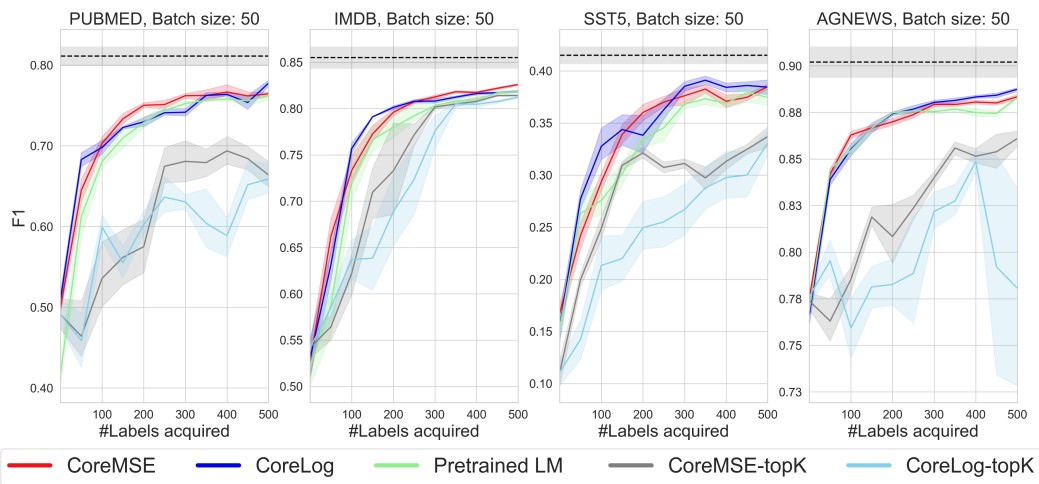

Figure 5: Learning curves of the model training with diversity. The dashline represents the performance of the backbone classifier trained on the entire dataset.

batch algorithms improve AL performance over the no-batch ones, and smaller batch sizes are more preferable to the large ones.

**Batch diversity** To further study the effectiveness of Algorithm 3, we considered the following variants: 1) Pretrained_LM: Instead of using the expected changes of scores to represent each unlabelled sample $x$, we used the embedding generated by the last layer of DistilBERT in $k$-Means clustering, which is similar to the BERT-KM in [40]; and 2) CoreMSE_top and CoreLog_top: We simply chose the top-$B$ samples ranked by $Q_x$. Figure 5 shows the results for all these variants. Our batch CoreMSE and CoreLog perform much better than the corresponding CoreMSE_top and CoreLog_top, which showcases Algorithm 3 can promote batch diversity that benefits AL for text classification. The performance difference between Pretrained_LM and batch CoreMSE/CoreLog indicates that representing each unlabelled sample as vector of expected changes in scores (i.e., Equation (6)) is effective in capturing the information to be used for diversification.

**Dynamic VS** We studied how the dynamic VS impacts the ensemble model training by comparing batch CoreMSE with dynamic VS to its following variations: 1) *3/30 epochs without VS*: ensemble model training without VS and each model was trained for 3 or 30 epochs [4], 2) *Fixed length (#1000) VS*: a pre-fixed validation set with 1000 labelled samples separate from the labelled pool, used in some existing AL empirical work; 3) *Constant VS*: a variant of dynamic VS where one random split was generated after each AL iteration and then shared by all the ensemble models. Figure 6 shows the learning curves of those compared AL methods. Dynamic VS gains an advantage after the third acquisition iteration on PUBMED and SST-5, the first iteration on AG NEWS. It is not surprising to see that 30 epochs without VS and Fixed length VS performs better in the early acquisition iterations,

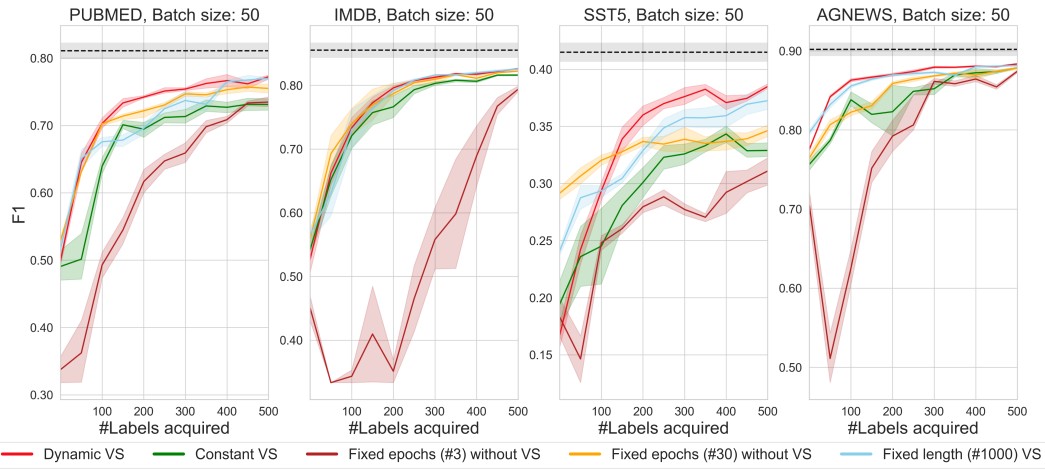

Figure 6: Learning curves of the model training with a dynamic validation set, constant validation set, fixed # epochs without validation set, fixed length # labels validation set for CoreMSE. The dashline represents the performance of the backbone classifier trained on the entire dataset.

since they used the whole augmented labelled pool in training DistilBERT, whereas CoreMSE used 70%. But choosing the number of epochs without a validation set is simply heuristic otherwise. Also Fixed length VS is midway between Constant VS and Dynamic VS, indicating the variability in ensembles inherent in the dynamic training sets is a source of improvement. More ablation results are reported in Appendix D.

## 5   Conclusion

We developed the BEMPS framework for acquisition functions for AL based around strictly proper scoring rules [9], or alternatively, Bregman divergences. These are fundamental scores in statistical learning theory and almost universally used when training neural networks, unlike simple errors, thus make a solid basis. In experiments we used mean squared error and log probability, making new acquisition functions CoreMSE and CoreLog respectively. For this, we developed convergence theory, borrowing techniques from [42], that we also extended to the earlier BALD acquisition function. A primary limitation of the theory is that it does not support the use of unlabelled data within the learning algorithm, for instance, as would be done by state of the art semi-supervised learning methods. Given that semi-supervised learning and AL really address a common problem, this represents an area for future work for our techniques. But note recent high performing batch AL uses semi-supervised learning, thus we excluded them from comparison.

For more efficient and effective evaluation of the new BEMPS based acquisition functions, we developed two techniques to improve performance, a batch AL strategy naturally complement to our BEMPS algorithms, and a dynamic validation-ensembling hybrid that generates high scoring ensembles but does not require a separate large labelled dataset. While this ensembling increases computational demand, it is only done for small datasets. These were both tested in isolation and shown to perform well. Though, interestingly, batch BEMPS works better than no-batch BEMPS, also seen for BatchBALD. This suggests that the one-step-look-ahead can be improved, and additional theory is needed. Finally, we followed some of the strong evaluation standards set in earlier research, but we kept to small labelled set sizes in keeping with text annotation practice, testing our approach against WMOCU, BADGE, BALD and baselines. Using mean squared error and log probability as the scoring rules yielded consistently high-performing AL on a variety of data sets.

## Acknowledgements

The work has been supported by the Tides Foundation through Grant 1904-57761, as part of the Google AI Impact Challenge, with Turning Point. Wray Buntine's work was also partly supported by DARPA's Learning with Less Labelling (LwLL) program under agreement FA8750-19-2-0501.

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
