# Diversity Enhanced Active Learning with Strictly Proper Scoring Rules: Appendix

**Wei Tan**
Monash University
wei.tan2@monash.edu

Lan Du*
Monash University
lan.du@monash.edu

Wray Buntine
Monash University
wray.buntine@monash.edu

## Abstract

This appendix contains further details regarding the theory and experiments. Section A gives the proofs of Lemma 1 about the properties of scoring rules and Theorem 1 about the convergence of active learning with (strictly) proper scoring functions. Section B provides details of the datasets, baseline implementation and experiment setup. Section C and Section D includes a comprehensive set of experimental results, including learning curves and pairwise comparison matrices on the four datasets that are not included in the main paper.

## A  Proofs

**Proof of Lemma 1** (Properties of scoring)

*Proof.* $Q_I(L) \geq 0$ by definition of entropy. Now an identity for entropy is that $\mathrm{I}\left(\Pr(A \mid B, C)\right) \leq \mathrm{I}\left(\Pr(A \mid B)\right)$. Which means given more evidence $C$, the conditional entropy of $A$ cannot increase. So the decrease in log volume is never negative. This means $\Delta Q_I(x|L) \geq 0$.

The result for $Q_S(L)$ follows directly by the definition of a scoring rule, because the expected proper score $\mathcal{E}_{\Pr(y|\theta,x)}[S(\Pr(\cdot \mid x), y)]$ has a minimum for any distribution $\Pr(\cdot \mid x)$ only when $\Pr(\cdot \mid x) = \Pr(y \mid \theta, x)$.

For $\Delta Q_S(x|L)$ we work as follows. Start with Equation (6) and reverse back in the scores:

$$
\mathcal{E}_{\Pr(y|L,x)}[\mathcal{E}_{\Pr(x')\Pr(y'|L,(x,y),x')}[S(\Pr(\cdot \mid L, (x,y), x'), y')]]
$$
$$
- \mathcal{E}_{\Pr(x')\Pr(y'|L,x')}[S(\Pr(\cdot \mid L, x'), y')]
$$
$$
= \mathcal{E}_{\Pr(y|L,x)}[\mathcal{E}_{\Pr(x')\Pr(y'|L,(x,y),x')}[S(\Pr(\cdot \mid L, (x,y), x'), y') - S(\Pr(\cdot \mid L, x'), y')]]
$$

where the second line is done by changing $\mathcal{E}_{\Pr(y'|L,x')}[\cdot]$ to $\mathcal{E}_{\Pr(y'|L,x,x')}[\cdot]$ (the model is fully conditional) then to $\mathcal{E}_{\Pr(y,y'|L,x,x')}[\cdot]$ and rearranging. The second line is $\geq 0$ due to the maximum properties of scoring functions used earlier. $\qquad \square$

**Proof of Theorem 1** (Convergence of active learning)

*Proof.* The proof for Lemma 5 in [5] can be readily adapted to show for $x$ occurring infinitely often in $L_n$ and $n \to \infty$ $\Delta Q_I(x|L_n)$ and $\Delta Q_S(x|L_n)$ both approach zero as $n \to \infty$ since $Q(L \cup \{(x,y)\}) \to Q(L)$ when $\Pr(\theta \mid L \cup \{(x,y)\}) \to \Pr(\theta \mid L)$. Then one adapts the proof of Theorem 1 in [5], which requires finiteness and discreteness of $x$, to show that $\Delta Q_I(x|L_n)$ and $\Delta Q_S(x|L_n)$ both approach zero as $n \to \infty$ for all $x$.

---

*Corresponding author

35th Conference on Neural Information Processing Systems (NeurIPS 2021).

Now consider $\Delta Q_I(x|L_n) = \mathcal{E}_{\Pr(\theta|L_n)}[\mathrm{KL}\left(\Pr(y \mid \theta, x) \parallel \Pr(y \mid L_n, x)\right)]$, by properties of the KL function. Let

$$\Theta_{NZ} = \left\{ \theta \, : \, \left( \lim_{n \to \infty} \Pr(\theta \mid L_n) \right) > 0 \right\}$$

Now $\theta_r \in \Theta_{NZ}$ due to the arguments of Lemma 5 [5]. From the KL approaching zero, it follows that for all $x$ and as $n \to \infty$, $\Pr(y \mid \theta, x)$ approaches $\Pr(y \mid L_n, x)$ for all $\theta \in \Theta_{NZ}$. This means that all $\theta \in \Theta_{NZ}$ yield identical $\Pr(y \mid \theta, x)$. Since the model is identifiable, $\Theta_{NZ}$ can only have one element, $\theta_r$.

Suppose $\Delta Q_S(x|L_n) \to 0$ as $n \to \infty$. Considering the final equation from the proof of Lemma 1, since $\mathcal{E}_{\Pr(y'|L_n,(x,y),x')}[S(\Pr(\cdot \mid L_n, (x,y), x'), y') - S(\Pr(\cdot \mid L_n, x'), y')] \geq 0$ from properties of the proper scoring rule, it follows that $\mathcal{E}_{\Pr(y|L_n,x)\Pr(y'|L_n,(x,y),x')}[S(\Pr(\cdot \mid L_n, (x,y), x'), y') - S(\Pr(\cdot \mid L_n, x'), y')]$ approaches 0 for all $x, x'$. Substitute the result from Savage about strictly proper scoring rules into the above simplification:

$$\mathcal{E}_{\Pr(y|L_n,x)\Pr(y'|L_n,(x,y),x')}[S(\Pr(\cdot \mid L_n, (x,y), x'), y') - S(\Pr(\cdot \mid L_n, x'), y')]$$
$$= \mathcal{E}_{\Pr(y|L_n,x)}[G(\Pr(\cdot \mid L_n, (x,y), x'))] - G(\Pr(\cdot \mid L_n, x'))$$

Because

$$\mathcal{E}_{\Pr(y|L_n,x)}[\Pr(y' \mid L_n, (x,y), x')] = \Pr(y' \mid L_n, x, x') = \Pr(y' \mid L_n, x')$$

(the second step because model is fully conditional) and $G(\cdot)$ is strictly convex, it must mean that $\Pr(y' \mid L_n, (x,y), x')$ approaches $\Pr(y' \mid L_n, x')$ for all $y, y'$ and $x, x'$ as $n \to \infty$. Now consider as $n \to \infty$

$$\mathcal{E}_{\Pr(\theta|L_n)}[\Pr(y \mid \theta, x)^2] = \Pr(y \mid L_n, (x,y), x)\Pr(y \mid L_n, x) \to \Pr(y \mid L_n, x)^2$$

and thus the variance of $\Pr(y \mid \theta, x)$ w.r.t. $\Pr(\theta \mid L_n)$ approaches zero for all $x, y$. Therefore, $\lim_{n \to \infty} \Pr(\theta \mid L_n)$ must be non-zero on a set of $\theta$ yielding identical $\Pr(y \mid \theta, x) = \lim_{n \to \infty} \Pr(y \mid L_n, x)$. The result follows using the logic above for $\Delta Q_I(x|L_n)$.

$\square$

## B  Detailed Experiment settings

### B.1  Datasets

We conducted experiments on a variety of binary and multi-class classification datasets, covering three distinct domains, i.e., News, medical abstracts and reviews. The basic information of those datasets is summarized in Table 1. IMDB is a balanced binary class dataset that includes both Positive and Negative classes. PUBMED and SST-5 are imbalanced datasets with multiple classes. PUBMED includes the five classes, which are Objective (O), Background(B), Conclusions (C), Results (R) and Methods (M), while SST-5 includes five rating scores from 1 to 5. AG NEWS is a large-scale balanced multi-class dataset, including four classes: Science/Technology (ST), World (W), Business (B) and Sports (S). In terms of text pre-processing, we used the tokenizer that comes with DisitlBert.

### B.2  Baseline implementation

Algorithm 1 shows the batch algorithm that we used to implemented all the AL methods considered in our batch experiments for a fair comparison. Note that the $acq.fn$ will return a score or a vector of scores for each unlabelled sample, which depended which AL method is used. To choose $B$ samples for a batch with Algorithm 1, we implemented each AL method as follows:

Table 1: The summary of dataset distribution

| Dataset | Imbalance | Multi-class | Domain | Label and size(%) |
|---|---|---|---|---|
| AG NEWS | No | Yes | News articles | ST/W/B/S(25/25/25/25) |
| PUBMED 20K RCT | Yes | Yes | Medical abstracts | O/B/C/R/M(8/12/15/32/33) |
| IMDB | No | No | Sentiment reviews | P/N(50/50) |
| SST-5 | Yes | Yes | Sentiment reviews | R1/R2/R3/R4/R5(13/26/19/27/15) |

**Algorithm 1** Deep Ensemble-based Active Learning with Dynamic Validation Set

---

**Require:** initial unlabelled data $U$, initial labelled data $L$,
**Require:** model/network ensemble $\Theta^E = \{\theta_1, ..., \theta_E\}$ built from labelled data $L$,
**Require:** acquisition function $acq.fn$, acquire batch size $B$, the number of acquisition iteration $N$
 1: Initialize: $i = 0, L_0 \leftarrow L, U_0 \leftarrow U$
 2: **while** $i < N$ **do**
 3:     **while** $e < E$ **do**
 4:         Random split $L_i$ into a training set and a validation set
 5:         Train the ensemble models $\theta_{i,e}(e \in E)$ given the current labelled training set and validation set
 6:     Form ensemble model $\Theta_i = \{\theta_{i,1}, \theta_{i,2}, ..., \theta_{i,E}\}$
 7:     **for** $x \in U_i$ **do**
 8:         Compute $r_x \leftarrow acq.fn.(x, \Theta_i)$
 9:     **if** $acq.fn$ is Max-Entropy or BALD **then**
10:         Acquire $B$ samples via ranking $r_x$
11:     **if** $acq.fn$ is other ALs **then**
12:         Acquire $B$ samples via clustering $r_x$ with k-MEANS++ or k-MEAN
13:     $L_{i+1} \leftarrow L_i \cup B$
14:     $U_{i+1} \leftarrow U_i \setminus B$

---

Table 2: The settings of the initial estimation pool size X and Top fraction size T for WMOCU, MOCU, CoreMSE and CoreLog on the different dataset

| Dataset | WMOCU | MOCU | CoreMSE | CoreLog |
|---|---|---|---|---|
| AG NEWS | X=600, T=0.5 | X=600, T=0.5 | X=600, T=0.5 | X=600, T=0.5 |
| PUBMED 20K RCT | X=800, T=0.5 | X=800, T=0.5 | X=800, T=0.5 | X=800, T=0.5 |
| IMDB | X=800, T=0.5 | X=800, T=0.5 | X=800, T=0.5 | X=800, T=0.5 |
| SST-5 | X=800, T=0.5 | X=800, T=0.5 | X=800, T=0.5 | X=800, T=0.5 |

**CoreMSE & CoreLog**: Each sample in the unlabeled pool was represented as a vector of scores computed by Eq (6) in the main paper. We then used $k$-MEANS to generate $B$ clusters and selected from each cluster a sample that is closest to the centre of the cluster to form the batch.

**Random** We sampled $B$ samples uniformly at random from unlabelled pool.

**Max-Entropy**: We chose top $B$ samples with the highest entropy of the predictive distribution [3, 4].

**BALD**: Similar to Max-Entropy, we chose top $B$ samples with the maximum mutual information based on how well labelling those samples would improve the model parameters [2].

**MOCU & WMOCU**: Similar to how we generated a batch using CoreMSE and CoreLog, we represented each sample in the unlabeled as a vector of scores computed by Eq (5 & 10) in [5], and then chose $B$ samples using the clustering approach.

**BADGE** Following [1], we represented each sample as a gradient embedding generatd from a pretrained language model, i.e., DistilBert in our experiments. Then, $k$-MEANS++, as used by Ash et al. [1], was then adopted to generate $B$ Clusters, from each of which we chose the sample closet to the cluster mean.

**ALPS**: Different from BADGE, we followed Ash et al. [1] to generate surprisal embeddings from DistilBert as inputs to $k$-MEANS. Then, a similar approach is used to choose the B samples in a batch.

**Pretrained LM**: We further used the embeddings generated by the last layer of DistilBert to represent unlabelled samples in clustering.

### B.3 Experiment setup

Table 2 shows the hyperparameters we implemented in the experiments. We set up the same estimation pool size and top fraction for AL methods in the specific domain.

# C  Model performance: Learning curves and pairwise comparison matrices

## C.1  Active learning with batch size 1

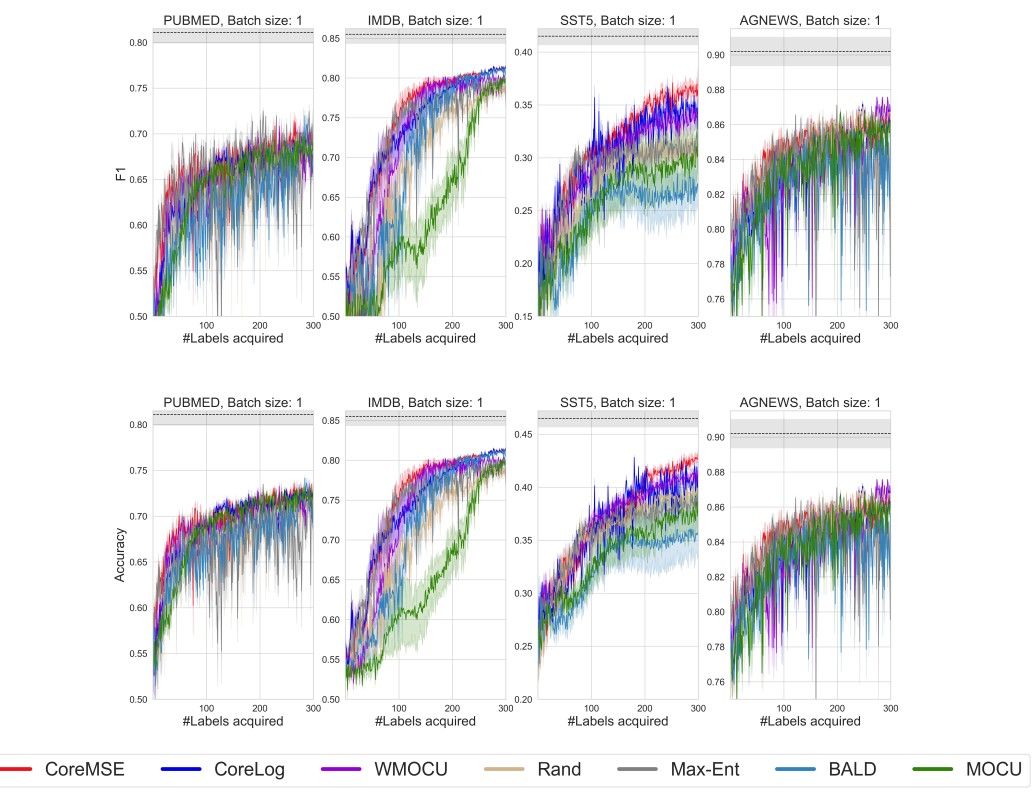

Figure 1: Learning curves on the four datasets, where we run all the AL methods with batch size 1. The dashline represents the performance of the backbone classifier trained on the entire dataset.

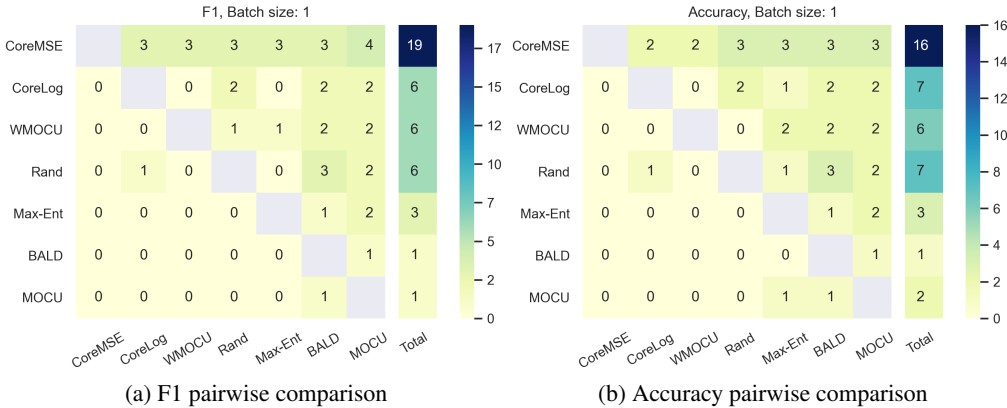

(a) F1 pairwise comparison  (b) Accuracy pairwise comparison

Figure 2: Pairwise comparison matrices of uncertainty active learning strategies combined on four datasets. Each number in the matrices represents the number of times the corresponding method in the row beats the method in the column. The maximum value is two based on the number of datasets tested. The number of the last column indicates the total winning times than the other methods. The higher value is better.

## C.2 Batch active learning

### C.2.1 Batch size 5

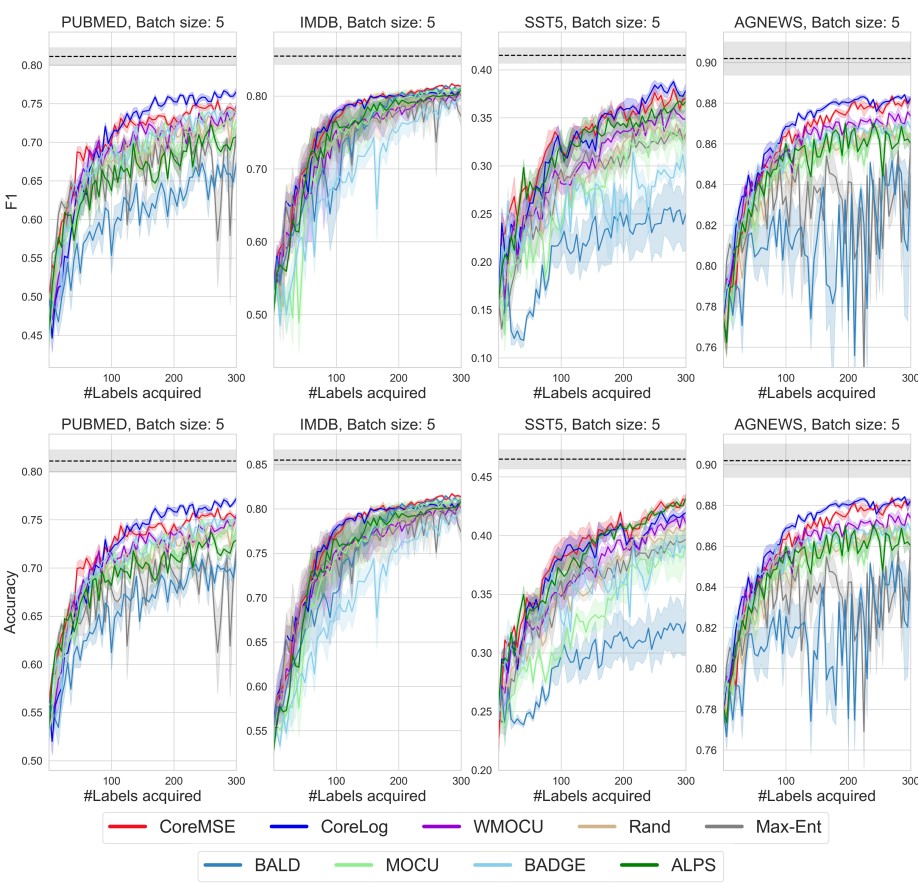

Figure 3: Learning curves of all the AL methods with batch size 5 on PUBMED, IMDB, SST-5 AND AG NEWS. The dashline represents the performance of the backbone classifier trained on the entire dataset.

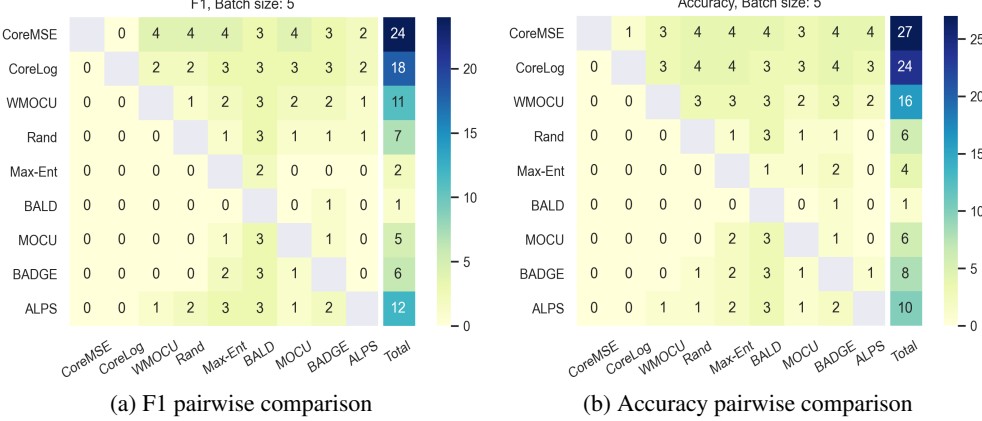

(a) F1 pairwise comparison

(b) Accuracy pairwise comparison

Figure 4: Pairwise comparison matrices of diversity active learning strategies with batch size 5. Each number in the matrices represents the number of times the corresponding method in the row beats the method in the column. The maximum value is four based on the number of datasets tested. The number of the last column indicates the total winning times than the other methods. The higher value is better.

## C.2.2 Batch size 10

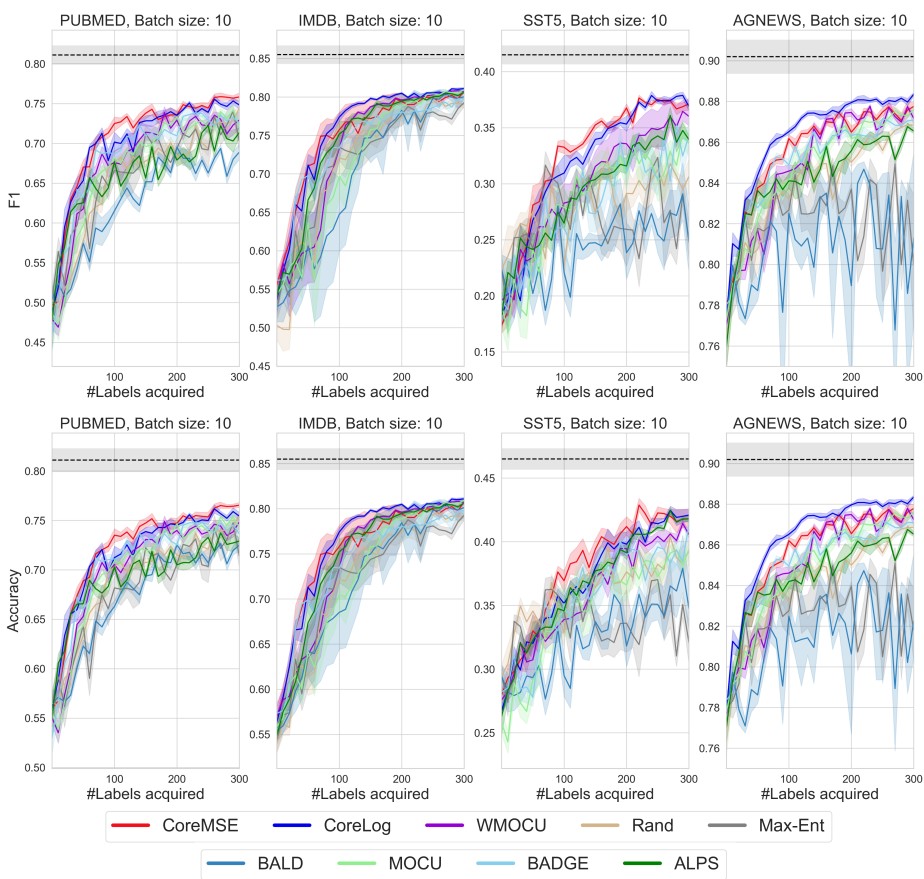

Figure 5: Learning curves of all the AL methods with batch size 10 on PUBMED, IMDB, SST-5 AND AG NEWS. The dashline represents the performance of the backbone classifier trained on the entire dataset.

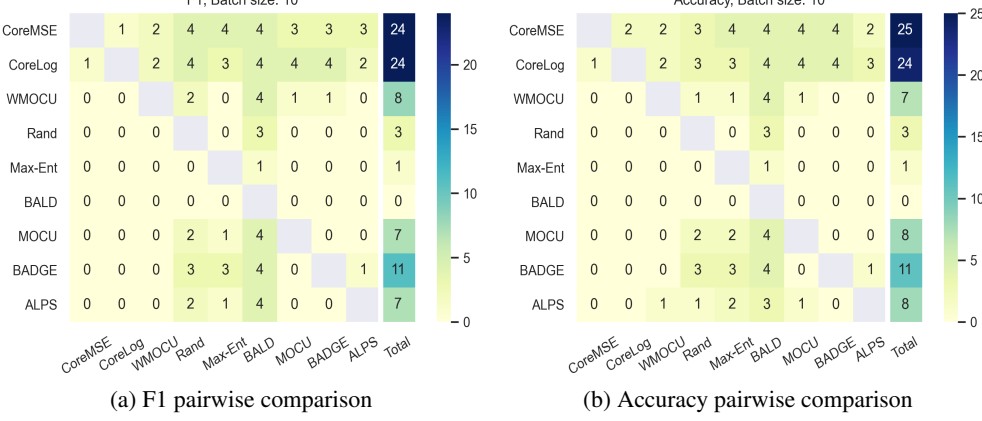

(a) F1 pairwise comparison        (b) Accuracy pairwise comparison

Figure 6: Pairwise comparison matrices of diversity active learning strategies with the batch size 10. Each number in the matrices represents the number of times the corresponding method in the row beats the method in the column. The maximum value is four based on the number of datasets tested. The number of the last column indicates the total winning times than the other methods. The higher value is better.

### C.2.3 Batch size 50

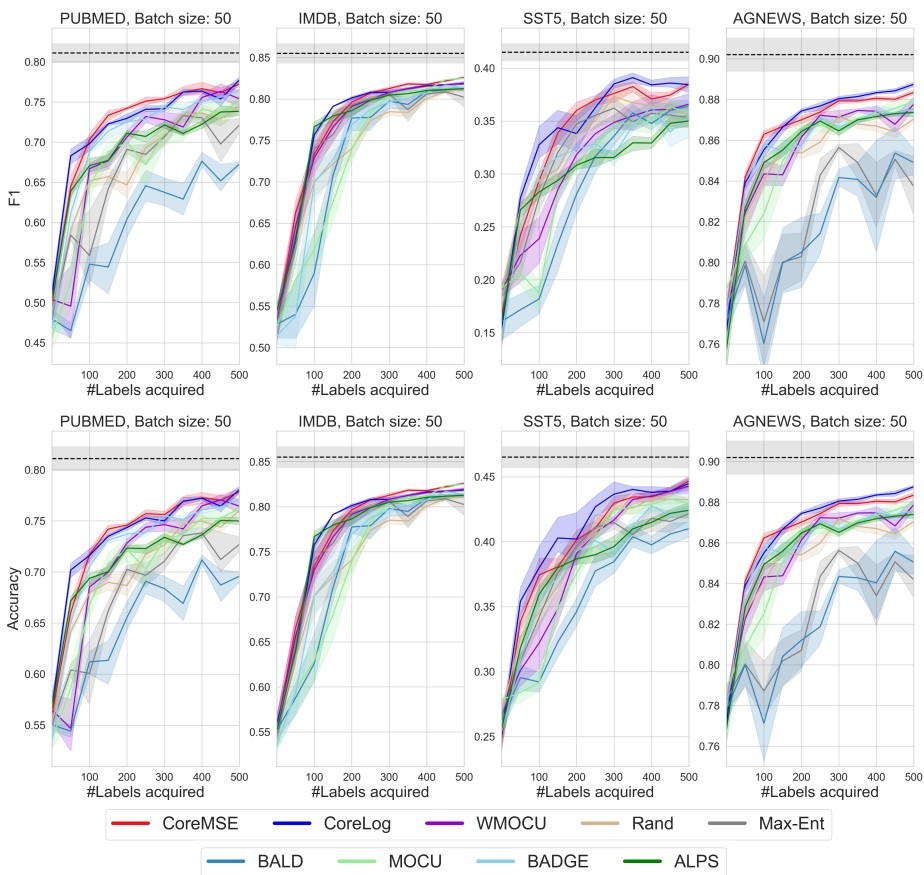

Figure 7: Learning curves of all the AL methods with batch size 50 on PUBMED, IMDB, SST-5 AND AG NEWS. The dashline represents the performance of the backbone classifier trained on the entire dataset.

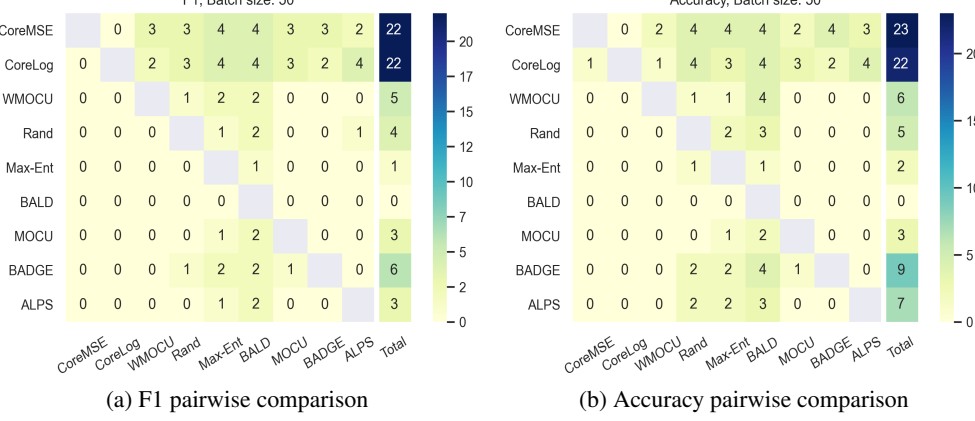

(a) F1 pairwise comparison        (b) Accuracy pairwise comparison

Figure 8: Pairwise comparison matrices of diversity active learning strategies with the batch size 50. Each number in the matrices represents the number of times the corresponding method in the row beats the method in the column. The maximum value is four based on the number of datasets tested. The number of the last column indicates the total winning times than the other methods. The higher value is better.

## C.2.4 Batch size 100

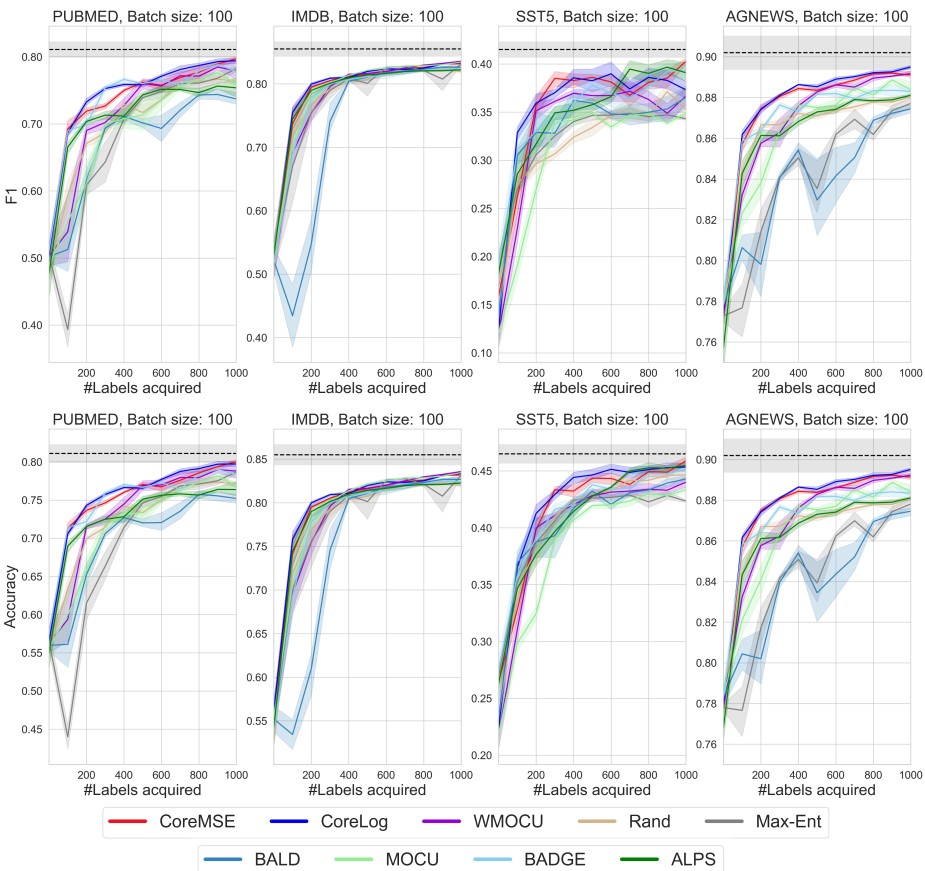

Figure 9: Learning curves of all the AL methods with batch size 100 on PUBMED, IMDB, SST-5 AND AG NEWS. The dashline represents the performance of the backbone classifier trained on the entire dataset.

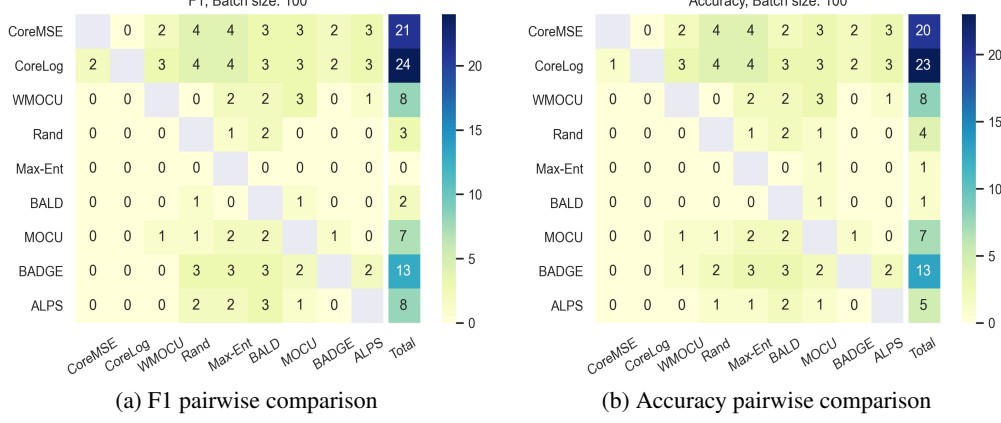

(a) F1 pairwise comparison

(b) Accuracy pairwise comparison

Figure 10: Pairwise comparison matrices of diversity active learning strategies with the batch size 100. Each number in the matrices represents the number of times the corresponding method in the row beats the method in the column. The maximum value is four based on the number of datasets tested. The number of the last column indicates the total winning times than the other methods. The higher value is better.

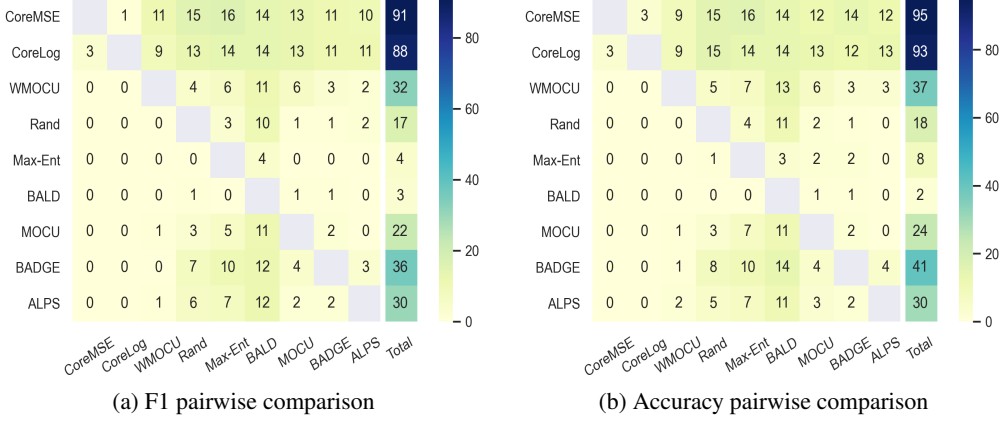

(a) F1 pairwise comparison

(b) Accuracy pairwise comparison

Figure 11: Pairwise comparison matrices of diversity active learning strategies with batch size 5, 10, 50 and 100. Each number in the matrices represents the number of times the corresponding method in the row beats the method in the column. The maximum value is four based on the number of datasets tested. The number of the last column indicates the total winning times than the other methods. The higher value is better.

# D  Ablation studies

## D.1  Batch size

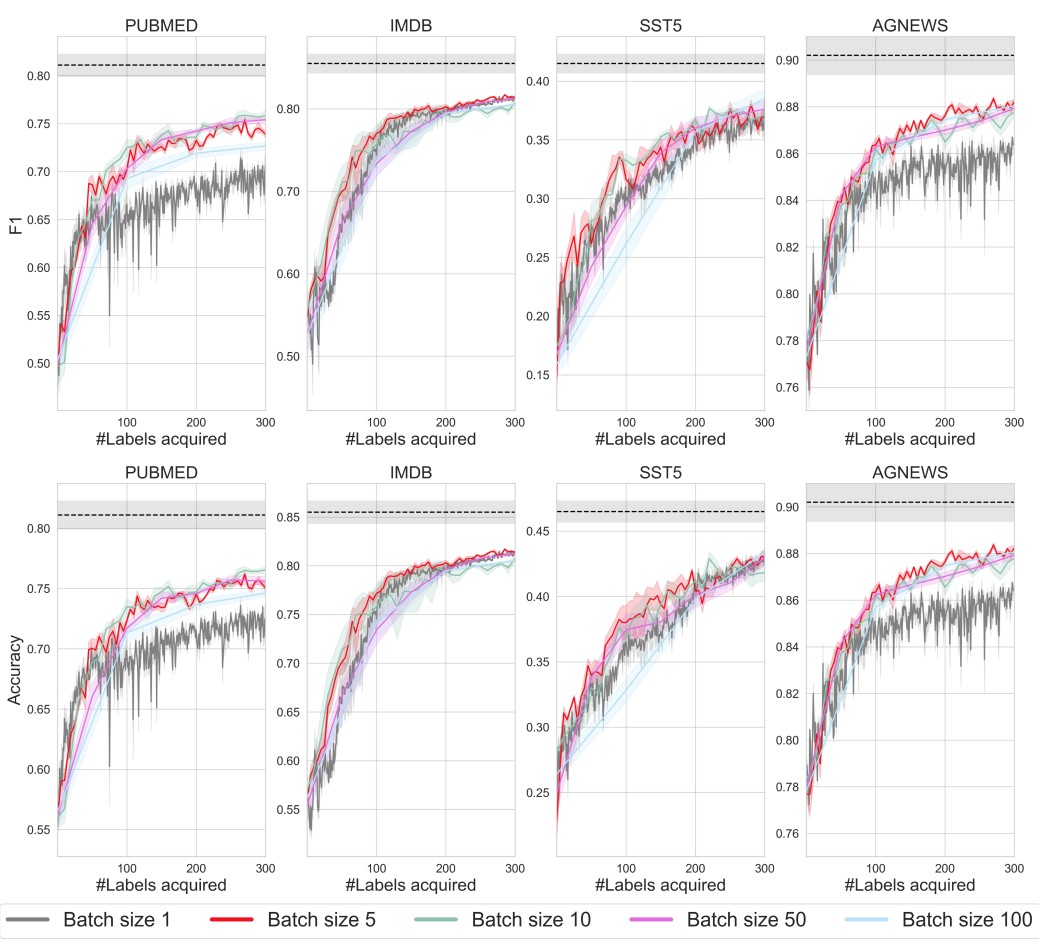

Figure 12: Learning curves of batch size 1, 5, 10, 50 and 100 for CoreMSE. The dashline represents the performance of the backbone classifier trained on the entire dataset.

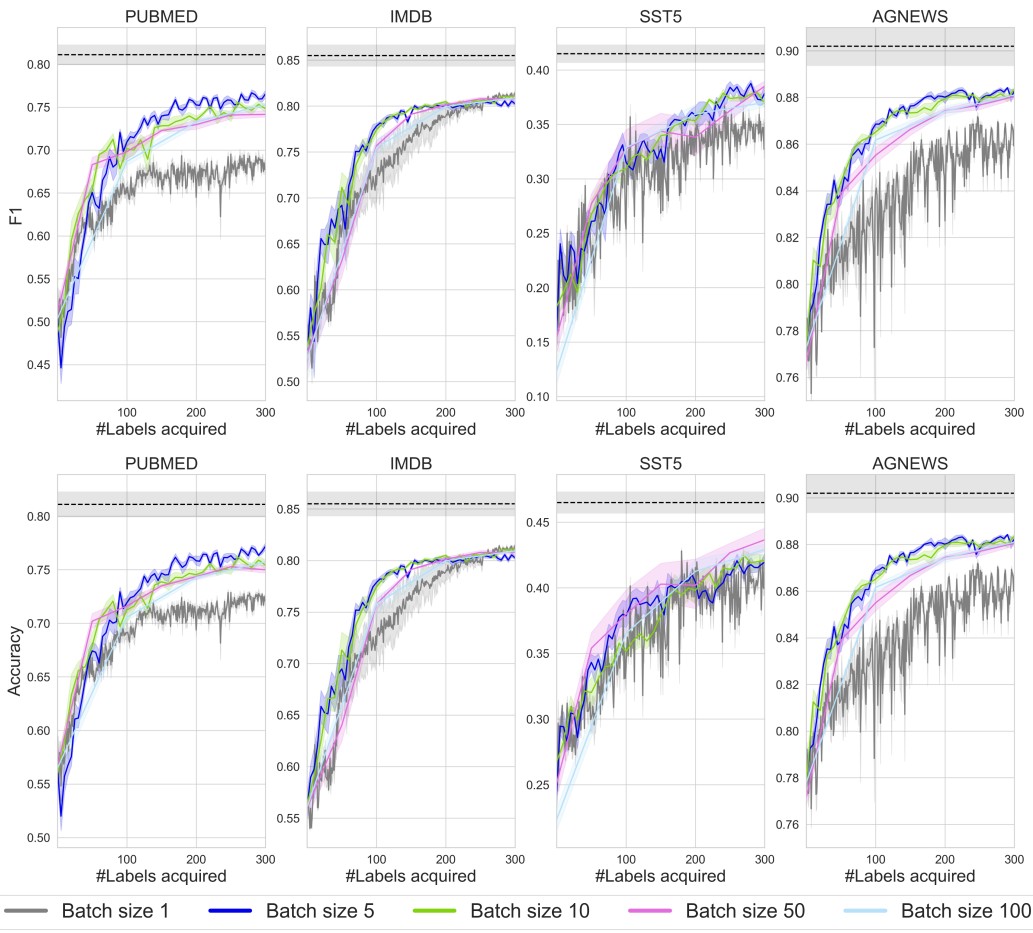

Figure 13: Learning curves of batch size 1, 5, 10, 50 and 100 for CoreLog. The dashline represents the performance of the backbone classifier trained on the entire dataset.

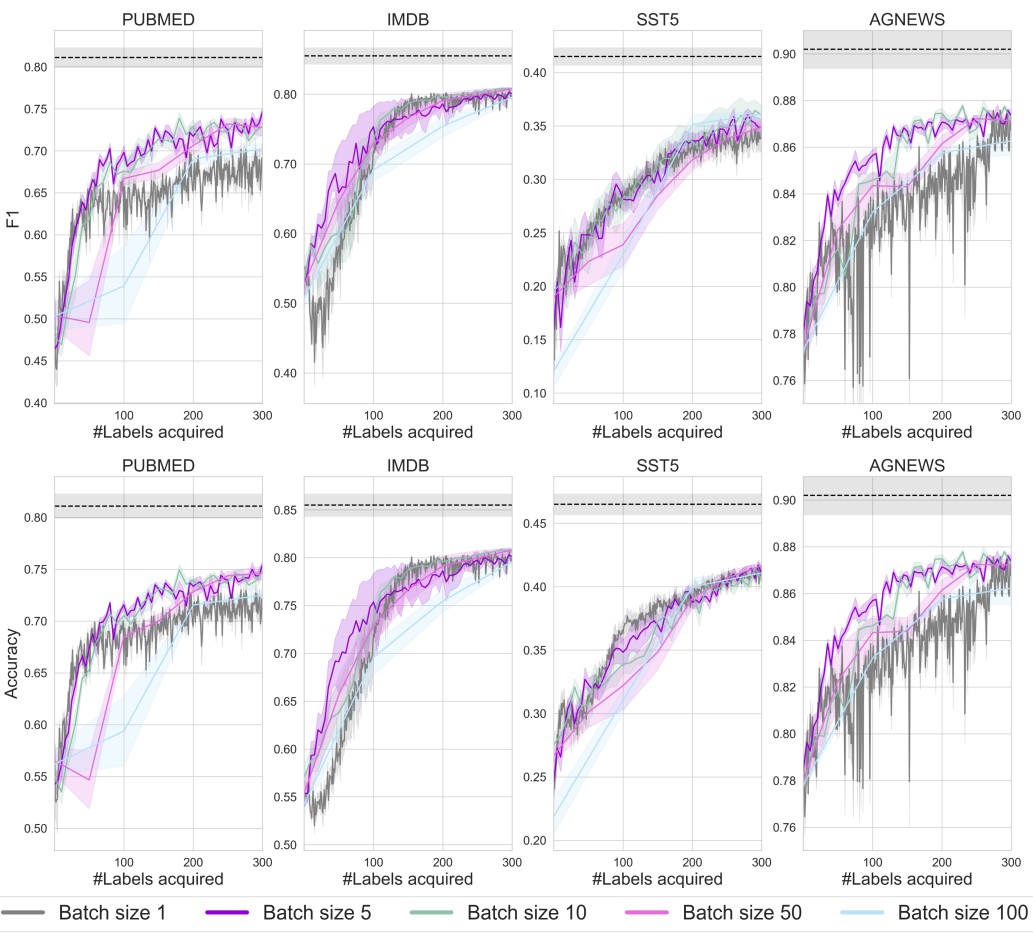

Figure 14: Learning curves of batch size 1, 5, 10, 50 and 100 for WMOCU. The dashline represents the performance of the backbone classifier trained on the entire dataset.

## D.2 Dynamic VS

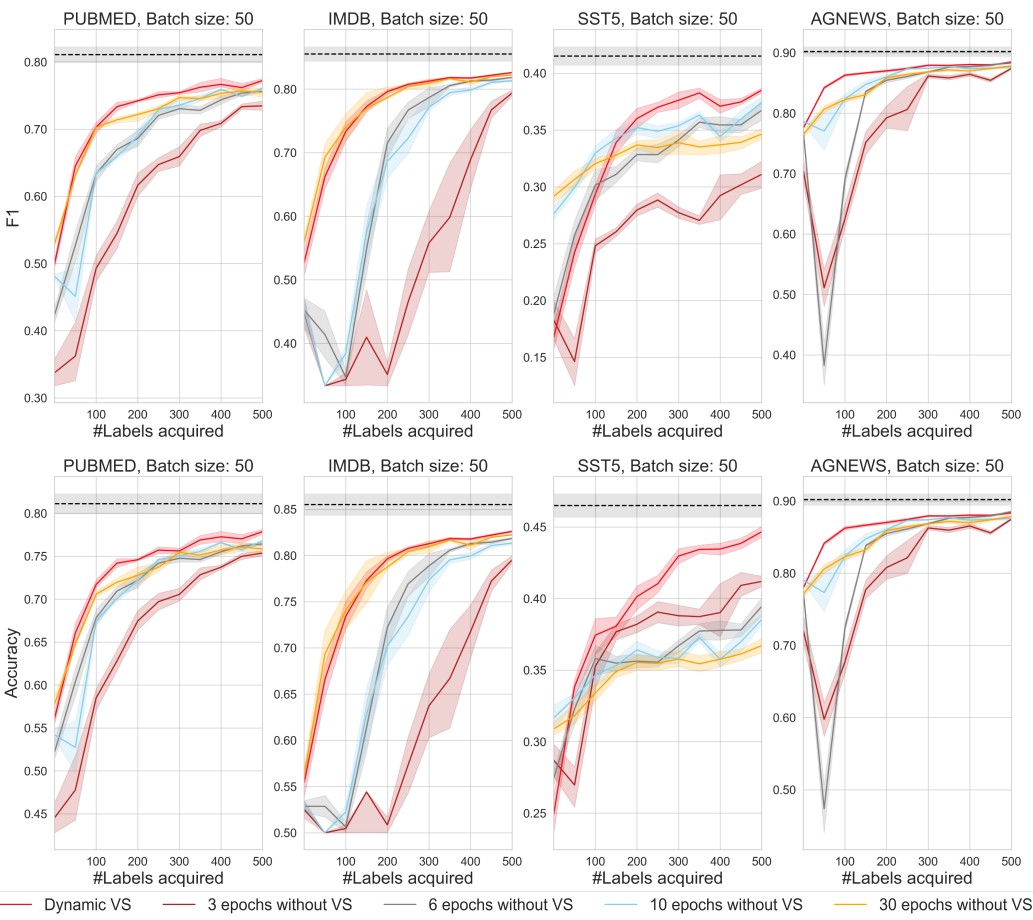

Figure 15: Learning curves of model training with dynamic validation set and fix number of epochs without validation set for CoreMSE. The dashline represents the performance of the backbone classifier trained on the entire dataset.