# OpenReview forum: "Diversity Enhanced Active Learning with Strictly Proper Scoring Rules"
_NeurIPS.cc/2021/Conference — NeurIPS 2021 Poster_

### Official Review · Reviewer_MYxD · 2021-07-06

**Rating:** 6
**Confidence:** 3

**Summary:**

The paper proposes new acquisition functions for Bayesian active learning, which replaces the classification errors in MOCU with strictly proper scoring rules. The paper proves the convergence of the new acquisition functions and demonstrates the AL performance with text classification benchmark datasets.

**Limitations And Societal Impact:**

I think the paper lacks analysis and discussions of the proposed methods. The paper do prove the convergence of the acquisition functions based on strictly proper scoring rules, but there are no further discussions on the sampling efficiency, and the guideline for choosing the strictly proper scoring rules (or the G functions). Although the paper provides sufficient experiments, the paper doesn't provide enough discussions on the experiment results.

I personally want to see the discussions on questions such as: since BALD/WMOCU are guaranteed to converge, why we still need to propose new methods? Why do we choose logarithmic scoring rule and squared error scoring rule? And based on the properties of the different scoring rules, in what conditions (large or small uncertainty, different sampling budgets) they can be more efficient than other competing methods?



**Main Review:**

 The paper proposes a class of acquisition functions based on strictly proper scoring rules, which are guaranteed to converge to the true model. The batch AL algorithm is concise and general. The experiments do demonstrate the effectiveness of the proposed methods.

**Time Spent Reviewing:**

3

---

> ### Author Response · Authors · 2021-08-09
> **The author's explanatory response to the reviewer's concerns**
>
> Thanks very much for your review, and valuable comments regarding our manuscript. We acknowledge the references that the reviewer pointed out and will add them into the next version of the paper. We will also consider the reviewer’s suggestion of using an running example and carefully proof-read our manuscript.
>
> * R4.1 Sampling efficiency is beyond the scope of this initial work, usually requiring very different theoretical methods because data is not IID (e.g., see Farquhar et al. ICLR 2021, “On Statistical Bias In Active Learning: How and When to Fix It”). Guideline for choosing the strictly proper scoring rules are discussed next.
>
> * R4.2 There are a number of questions here. First, as said in response to R2.2 BALD is asking the wrong question, how to reduce uncertainty on model parameters, as demonstrated by its poor performance. So why would you need a whole family of rules over and above WMOCU? Great question, and one we wrongly assumed others would understand, so didn’t address. The decision theory community provides some guidance here, e.g. “Choosing a Strictly Proper Scoring Rule,” E.C. Merkle & M. Steyvers 2013 and the statistical community (e.g., Gneiting, T., & Raftery, A. E, 2007) also gives examples for different inference tasks [10]. For instance, one can adapt a scoring rule depending on whether a particular set of utilities or recall is to be used. One could also be Bayesian about it and stick with a log probability but this presupposes a particular model family. In most cases one can create a particular model to match just about any Bregman divergence (e.g., minimum squared errors is a Gaussian). In practice we can use robust models (e.g., a Dirichlet-multinomial rather than a multinomial, a negative binomial rather than a Poisson, Cauchy rather than Gaussian) in our log probability. So being Bayesian suggests using a robust model with a log probability score.
>
>   Thus, this shows a wide variety of inference tasks can be addressed by our approach, as a well as providing a framework for Bayesians. This goes way beyond simple minimum errors of WMOCU.
>
>   Now for the particular cases we used, log probability versus Brier score, minimising Brier score gets the probability right in a least squares sense, paying less attention to very low probability events, whereas log probability gets the probability scales right, paying attention to all events. Brier score is known to work well with minimum errors tasks and non-extreme utility tasks. I hope from this example you can see we can reason about the proper score and from there map one to a desired inference task.
>
>   Again, great motivating question and needs explaining in the paper.
>
> References:
> * Sebastian Farquhar, Yarin Gal, and Tom Rainforth. On statistical bias in active learning: How and when to fix it, 2021
> * Edgar C. Merkle & Mark Steyvers, 2013. "Choosing a Strictly Proper Scoring Rule," Decision Analysis, INFORMS, vol. 10(4), pages 292-304, December.

---

### Official Review · Reviewer_eoGS · 2021-07-14

**Rating:** 6
**Confidence:** 4

**Summary:**

The paper addresses the problem of active learning by querying examples that will lead to the most reduction in classification
error, which is a fairly common idea. It proposes to place this in the
context of strictly proper scoring rules in order to derive new
example acquisition functions and generalize existing
ones. Experiments on multiple datasets illustrate this for two common
scoring rules, with extensive analysis, showing that the proposed
methods perform well.


**Limitations And Societal Impact:**

Yes this is mentioned in the checklist, although not really relevant for this paper.


**Main Review:**

Plus:
+ clean framework with some theoretical guarantees
+ experimental comparison investigates many aspects of the method
+ consistent performance improvement over alternatives

Minus:
- semi-supervised learning is specifically excluded
- novelty seems limited
- many moving parts

The first part of the paper is a quick (and steep) intro of the theory behind the method. It is very formal and could benefit from more actual examples of instantiations of the concepts early on, but this is a matter of taste. This is followed by a sizeable experimental section which covers a number of questions and analyzes the impact of various aspects of the method (batch size and diversity, dynamic validation set). The results are convincing, with learning curves (with error bars) and statistical testing of improved performance.

The paper seemed technically correct, building upon previous work in machine (active) learning and statistics to place the method within a clean theoretical framework using strictly proper scoring rules and extending some existing methods in the process. Some of the contributions seem modest, for example the "dynamic validation" is essentially randomly splitting training and validation sets during ensemble training -- producing ensembles by samping train/val set has been used extensively and it is unclear whether there is something clearly novel in the particular way it is done here. There are many moving parts and hyper-parameters in the method. Some of them are tested in the ablation study in Sec. 4.2. Results suggest that the proposed method functions well, but for a fairly specific choice of these settings. One may wonder how these settings would generalize to other domains or data sets, for example with much larger datasets.

The main limitation of the paper imho is that it specifically excludes semi-supervised learning from consideration and comparisons (the "fully conditional" assumption). SSL and AL go hand-in-hand since at least co-training. Even though the nice theoretical framework does not accomodate SSL, it is hard not to at least compare with real state-of-the-art in AL.
[**EDIT** The authors' point is fair that one has to start somewhere]

Clarifications:

Is the 'running time per single acquisition' normalized for batch size, or for one batch (i.e. one run of Alg.3)? I would guess the latter, but 'single acquisition' introduces a bit of ambiguity.

It could help to place the sample sizes l^k_i,j,d on Fig 1 (left).

Minor points:

The pairwise comparison matrix is a poster case for correcting for multiple comparisons.

In the conclusion BEMPS seems to turn into BERPS.

Typo: generated (l.92)

**EDIT** Thanks for the detailed reply. Review is updated accordingly.

**Time Spent Reviewing:**

3.5

---

> ### Author Response · Authors · 2021-08-09
> **The author's explanatory response to the reviewer's concerns**
>
> Thanks very much for your review, and valuable comments regarding our manuscript. We acknowledge the references that the reviewer pointed out and will add them into the next version of the paper. We will also consider the reviewer’s suggestion of using an running example and carefully proof-read our manuscript.
>
> * R3.1 The dynamic validation and ensembling are listed as our second contribution in the paper, and indeed contribute sizably to the learning performance, as indicated in Figure 6. Experimentally, this is significant because it is neither practical nor realistic to assume there is a separate and fixed validation set available a priori in real world applications. The dynamic validation based on the acquired pool demonstrates such an assumption is not necessary. One might argue it is a simple approach but its importance is in demonstrating realism of the empirical work, too often previously ignored. The fundamental and more significant contribution is (1A) linking active learning to the half-century of statistical learning theory of strictly proper scoring rules and Bregman divergences (see responses to R4.2 and R1.2 for why this is good), together with (1B) the convergence proof, a non-trivial adaptation of the great proofs for WMOCU. This is highlighted by their consistently superior performance.
>
> * R3.2 There are two moving parts: the size of the validation set (X) and top fraction T, the proportion of the top-k data in Algorithm 3. The first is strictly a trade-off (bigger, more accuracy, better results but slower computation) so not an issue. The top fraction is a heuristic parameter so can only be investigated empirically, which we agree should be done. We ran a set of experiments where the size of X was fixed, and the value of T varied from 0.2 to 1. The results show that the performance of our model does not vary significantly when the value is greater than or equal to 0.5. We will attach the result to the supplementary materials.
>
> * R3.3 We viewed the exclusion of SSL as a deep theoretical insight. We agree entirely, SSL and AL go hand-in-hand, but we believe our technique advances the theory substantially for the non-SSL case, and see our response to R4.2. We know of no theoretically derived approaches combining AL and SSL, , and we felt the theory for AL needed a strong, direct connection with the half century of statistical learning theory (i.e., our approach) before launching onto the far harder theoretical question of AL + SSL. SSL presents very different theoretical requirements.
>
> * R3.4 The Reviewer is right. The average running time was calculated for one run of Algorithm 3.
>
> * R3.5 In Figure 1, we selected five samples from the entire iterations. The step size of l^k_i,j,d between the samples is 50. For example, the first sample l^1_i,j,d is selected at the 50th iteration, the second sample l^2_i,j,d is selected at the 100th iteration and etc. We will revise our manuscript accordingly.

---

### Official Review · Reviewer_eduq · 2021-07-17

**Rating:** 7
**Confidence:** 3

**Summary:**

The paper analyzes possible acquisition functions for active learning in the field of text classification. The authors introduce BEMPS (Bayesian Estimate of Mean Proper Scores), which leverages the existing Expected Loss Reduction method to estimate the increase in scores like log probability and negative mean square.

**Main Review:**

The authors propose a novel approach to an important problem: increasing the diversity of the examples selected for labeling during active learning. The paper is reasonable well written, though the prose is awkward at times (e.g., lines 33 and 44-48) and makes a compelling case for the proposed approach. The contributions appear to be original.

One way to improve the paper's readability would be to add an intuitive running example before you get to Section 3, which is quite dry and dense on the allotted two pages. You can probably do this in less than one page, based on on figure and a few paragraphs. Illustrating your three main contributions (lines 35-48) in an intuitive manner will dramatically improve the paper's readability for a general audience. You are trying to do this in section 3.2 in a more formal manner.

 In Fig 1:
- is the dashed line (in the image on the left) the performance if training on the entire dataset?
- could you please discuss why RAND outperforms BALD & MOCU  (and ties with Max Ent) in this scenario?

OTHER COMMENTS:
- given that the work on WAAL, VAAL, and MAL is on image datasets, you do not have to make three times the argument on why you are not comparing with them (one time is enough :)
- however, the idea of combining active & semi-supervised learning is not new - see [McCallum & Nigam, 1998] or [Muslea at at, 2002]
- line 32: newer --> recent
- line 44:  wth --> with             (please spell-check!)

McCallum, A., & Nigam, K. (1998). Employing EM in pool-based active learning for text classification. In ICML (pp. 359–367)

Muslea, I., Minton, S., & Knoblock, C. A. (2002, July). Active+ semi-supervised learning= robust multi-view learning. In ICML (Vol. 2, pp. 435-442).

**Time Spent Reviewing:**

2.5

---

> ### Author Response · Authors · 2021-08-09
> **The author's explanatory response to the reviewer's concerns**
>
> Thanks very much for your review, and valuable comments regarding our manuscript. We acknowledge the references that the reviewer pointed out and will add them into the next version of the paper. We will also consider the reviewer’s suggestion of using an running example and carefully proof-read our manuscript.
>
> * R2.1 The reviewer is right. The dashed line represents the performance of the backbone classifier trained on the entire dataset, severing as the performance upper bound.
>
> * R2.2 BALD uses mutual information to score samples based on how their labels could inform the true model parameter distribution, which will be problematic if the uncertainty of model parameters has a reduced relationship to the classification performance; and MOCU has convergence issues as ELR, as pointed out by [43]. One should not be surprised that RAND beats them. Many recent papers using BALD as a comparison algorithm also exhibit this phenomena [13, 47].
>
> References
>
> * [47] Yilun Zhou, Adithya Renduchintala, Xian Li, Sida Wang, Yashar Mehdad, and Asish Ghoshal. Towards understanding the behaviors of optimal deep active learning algorithms. In Arindam Banerjee and Kenji Fukumizu, editors, Proceedings of The 24th International Conference on Artificial Intelligence and Statistics, volume 130 of Proceedings of Machine Learning Research, pages 1486–1494. PMLR, 13–15 Apr 2021.

---

> ### Comment · Reviewer_eduq · 2021-08-30
> **Original rating (ACCEPT) is maintained after reviewing the authors' comments**
>
> I would like to thank the authors for properly addressing the reviewers' comments and questions. In this reviewer's opinion, the paper still is a solid ACCEPT.

---

### Official Review · Reviewer_P5cX · 2021-07-20

**Rating:** 7
**Confidence:** 5

**Summary:**

This paper proposes a novel active learning method - called BEMPS (Bayesian Estimate of Mean Proper Scores) - in the context of text classification.
Based on the ELR (expected loss reduction) approach, the authors propose to estimate the increase in the so-called "strictly proper scores"  such as the log probability or negative mean square error.
They show that the proposed method, BEMPS, makes the classifier asymptotically converge to the optimal classifier as the active learning process continues.
For proof, it builds upon techniques that were recently utilized for proving the convergence of the active learning scheme based on weighted MOCU (mean objective cost of uncertainty).
Based on extensive evaluations on various benchmarks, this study shows that BEMPS generally outperforms other state-of-the-art active learning methods.



**Limitations And Societal Impact:**

Some limitations of the proposed method are discussed in the conclusion.
Potential negative societal impact is not discussed in detail.



**Main Review:**

GENERAL COMMENTS

Overall, this is an interesting and well-written paper that improves the current state-of-the-art in active learning (AL) - especially in the context of text classification.
The paper presents strong performance evaluation results that clearly demonstrate the advantages of the proposed method, BEMPS, while also presenting interesting convergence proofs of the proposed active learner.
Although the convergence proofs are based on techniques that were recently used to show the convergence of the WMOCU (weighted MOCU)-based active learning scheme, the authors show that the proposed approach often yields favorable learning outcomes compared to the WMOCU-based active learning scheme as well as several other AL algorithms that were tested.


DETAILED COMMENTS

1. What is the main impact of changing the ELR scheme to utilize proper scoring rules or the Bregman divergence instead of the classification error?
Considering that the main goal of the active learning process is to enhance the performance of the classifier (therefore reducing its classification accuracy), would this change lead to any "semantic" differences of the learning process compared to other ELR methods?
Although convergence to the optimal classifier is shown in the paper, what would be the main benefits of changing the scoring scheme?
Providing further insights would be useful.
Especially, considering that the original WMOCU-based scheme, to which BEMPS is compared, takes an "objective-based" approach, which focuses on the expected reduction of the classification error, it would be meaningful to compare BEMPS with WMOCU from this perspective.

2. Based on the presentation in Sec. 3.1., what is the relation between  BEMPS and WMOCU?
Does the WMOCU scheme become a special case of BEMPS (or vice versa) under certain conditions?

3. What do"fully conditional model" and "full conditionality" mean?
Please provide a clear definition.

4. It is unclear how MOCU was utilized in the active learning framework.
For example, in order to evaluate the expected gain of a new data point (i.e., label) acquisition, expected conditional MOCU given the new data should be evaluated.
But as the current paper doesn't provide the relevant details, it is not clear how MOCU was utilized.

5. Further discussion is needed regarding the relationship between the batch size and  the overall performance (e.g., lines 266-267)
For example, why would the no-batch case perform worse compared to using small batches?
And would a larger batch size lead to better performance as the number of acquired labels increase outperforming small batch cases? (Although not initially, when the number of newly acquired data points is small?)

6. Please carefully proofread the manuscript and the supplementary material to correct any typos/errors.
- a prior -> a priori (line 110)
- equation # missing in the supplementary material (line 68)

-----------

Note:

I acknowledge that the authors' response to the original review comments has been reviewed and considered in the rating.



**Time Spent Reviewing:**

12

---

> ### Author Response · Authors · 2021-08-09
> **The author's explanatory response to the reviewer's concerns**
>
> Thanks very much for your review, and valuable comments regarding our manuscript. We acknowledge the references that the reviewer pointed out and will add them into the next version of the paper. We will also consider the reviewer’s suggestion of using an running example and carefully proof-read our manuscript.
>
> * R1.1 Consider fitting a neural net. People almost universally use a differentiable scoring rule, and mostly they use strictly proper scoring rules. Errors are rarely used. WMOCU was an adaptation to MOCU to get around the fact that MOCU, based on errors, was shown to have flaws as an acquisition function. This comes as no surprise to people familiar with the theory of strictly proper scoring rules. With BEMPS we took another approach to make the convergence theory work, generalising expected errors to expected score. Our problem with WMOCU is that it only applies to errors. In our medical applications errors are less irrelevant when doing an inference task; our applications often have tradeoffs between precision and recall or a focus on a subclass of error. Using strictly proper scoring rules means your classifier becomes calibrated, and then it can be used for any kind of inference task (different utilities, precision-recall tradeoffs, etc.). Moreover, as highlighted in the response to Review 4 (See R4.2), the scoring rule can also be adapted to the particular inference task.
>
> * R1.2 With a fully conditional model, the posterior of θ is unaffected by unlabelled data, which means p(θ|L, U) = p(θ|L) for any unlabelled data U (this is given at line 100 in the paper). Then “full conditionality” holds when a model is fully conditional.
>
> * R1.3 Both WMOCU and MOCU were implemented based on the algorithm described in the weighted MOCU paper [43]. Yes, the reviewer is correct, one will need to compute the expected conditional MOCU given the new acquired sample. MOCU-based acquisition function computes the reduction of MOCU in a similar manner to ELR, see Section 3.1 of [43].
>
> * R1.4 This is a fundamentally important question, and we surmise it is due to the sub-optimality of repeated one-step lookahead optimising acquisitions. The no-batch case is based on doing 5 one-step lookahead optimising acquisitions (Algorithm 1 and Algorithm 2) whereas the
> batch case does 5 acquisitions at once, heuristically, incorporating a form of diversity based directly on the error surface (Algorithm 1 and Algorithm 3). BatchBALD is an inspiration here because they show how to work with a batch, but also BatchBALD exhibited the same
> property. We viewed answering this question by implementing BatchBEMPS beyond the scope of this initial paper.

---

### Author Response · Authors · 2021-08-09
**Thank you all for your reviews**

We want to express our deepest gratitude to all reviewers for their time and effort in our work. We were pleased to see that reviewers suggested the paper provided valuable insights and relevant analysis. They also made excellent recommendations for further improving the paper, which we address in the comments section below and will upload in the next version.

---

### Decision · Program_Chairs · 2021-09-27

**Decision:**

Accept (Poster)

**Comment:**

The paper develops a new acquisition function for active learning, which is referred to as Bayesian Estimate of Mean Proper Scores or BEMPS for short. BEMPS generalizes weighted Mean Objective Cost of Uncertainty or WMOCU, a recently developed active learning acquisition function based on expected loss reduction (ELR) by using the strictly proper scoring rules to replace the classification errors. This generalization offers a more natural and principled way to avoid the acquisition function from getting stuck before reaching the optimal classifier, while also ensuring a broader applicability where errors are less informative.

All the reviewers agree that this is a well-written paper, which tackles an important and interesting problem. They also acknowledge the theoretical contributions, including the proof of convergence to the true model parameter (but guarantee on sample efficiency is not provided, which is important for active learning). Evaluation on text classification tasks shows convincing results as compared with some recent baselines.

Most questions raised by reviewers have been adequately addressed by the authors during the discussion phase. Given the similarity between the proposed BEMPS and existing ELR based acquisition functions (some of which are also proved to converge to the optimal classifier), it is important for the authors to include some additional insights on the advantage of using the strictly proper scoring rules, for example, on what specific conditions that BEMPS can bring additional benefit to the active sampling process, in the final version of the paper.